# Formation of the Atlantic Meridional Overturning Circulation lower limb is critically dependent on Atlantic-Arctic mixing

Dipanjan Dey [1,2] ✉, Robert Marsh[1], Sybren Drijfhout[1,3], Simon A. Josey [4], Bablu Sinha[4], Jeremy Grist [4] & Kristofer Döös [5]

Deep-water formation in the eastern Subpolar North Atlantic Ocean (eSPNA) and Nordic Seas is crucial for maintaining the lower limb of the Atlantic Meridional Overturning Circulation (AMOC), of consequence for global climate. However, it is still uncertain which processes determine the deep-water formation and how much Atlantic and Arctic waters respectively contribute to the lower limb. To address this, here we used Lagrangian trajectories to diagnose a global eddy-resolving ocean model that agrees well with recent observations highlighting the eSPNA as a primary source of the AMOC lower limb. Comprised of 72% Atlantic waters and 28% Arctic waters, the density and depth of the AMOC lower limb is critically dependent on Atlantic-Arctic mixing, primarily in the vicinity of Denmark Strait. In contrast, Atlantic waters gaining density through air-sea interaction along the eastern periphery of Nordic Seas and not entering the Arctic Ocean make a negligible contribution to the lower limb.

The Atlantic Meridional Overturning Circulation (AMOC) plays a key role in the global climate. The near-surface branch of the AMOC (upper limb) transports heat and salt northwards from the Equator, which helps to offset the atmospheric cooling at mid-latitudes (particularly central Europe, Scandinavia), moderating regional climate[1,2]. At high-latitudes, colder, fresher, and denser water-masses (lower limb of AMOC) are formed that travel equatorward via deep ocean currents[3,4]. A continuous sinking branch of dense waters is thus necessary to maintain the northward flow of warm and buoyant water in the Atlantic Ocean. Paleoclimate records reveal a strong link between the strength and location of dense water formation and abrupt temperature fluctuations around the globe over the last glacial cycle[5]. Additionally, modelling studies suggest that a shut-down of dense water formation could be a potential tipping point in future climate[6]. Given its importance, it is crucial to understand how and where the water of the lower limb of the AMOC is formed. A trans-basin observational array (OSNAP) showed that the lower limb preferably forms north of the OSNAP-East section, which extends from the south-eastern tip of Greenland to Scotland[7,8]. The strength of the MOC at the OSNAP-East array between August 2014 and May 2018 was estimated at 16.8 ± 0.6 Sv (1 Sv ≡ $10^6$ m$^3$ s$^{-1}$) and the density surface that separates the lower and upper limb was 27.55 kg m$^{-3}$ [8]. The source of the lower limb waters and the processes leading to its formation are still debated, however. Hypotheses range from open ocean convection in the Greenland and Iceland Seas to densification of the Atlantic waters at the Arctic shelves and along the eastern periphery of the Nordic Seas[9,10]. The annual dense water production rate in the Greenland and Iceland Seas was observed to be $O$(1 Sv)[11–13], much lower than total volume flux of the lower limb of the AMOC, which ranges from 15 Sv to 18 Sv[8,14–16]. Dense waters from the Nordic Seas are transported across the Greenland-

[1]School of Ocean and Earth Science, University of Southampton, Southampton, UK. [2]School of Earth, Ocean and Climate Sciences, Indian Institute of Technology Bhubaneswar, Bhubaneswar, India. [3]Royal Netherlands Meteorological Institute, De Bilt, Netherlands. [4]National Oceanography Centre, Southampton, UK. [5]Department of Meteorology, Stockholm University, Stockholm, Sweden. ✉e-mail: d.dey@soton.ac.uk

Scotland Ridge (GSR) to the eSPNA primarily in the East Greenland Current (EGC)[10,17], with a smaller contribution via the Iceland-Faroe Slope Jet[18]. The transport of dense waters (which also include overflow waters that are known to have potential densities >27.8 kg m$^{-3}$) across the GSR has been estimated to be $6.6 \pm 0.4$ Sv[19], and explaining approximately half of the total observed AMOC lower limb[8,15]. The other half was found to be associated with oceanic heat loss in the Irminger Sea and Iceland basin[19].

However, the role of interior mixing in forming the AMOC lower limb is still unclear and could be large, given observational and modelling evidence for vigorous mixing in the Denmark Strait[20,21]. A recent study reported that air-sea flux cannot fully explain the time-mean AMOC magnitude at OSNAP-East and a residual mixing is necessary to explain the gap[22]. Additionally, it is not well known how much of the total AMOC strength could be attributed to Atlantic and Arctic waters. It is important to note that the Atlantic and Arctic water denote water parcels that are arriving from the Atlantic and Arctic Ocean after upstream property modification. To answer these questions, we calculated and diagnosed Lagrangian trajectories for the present-day climate using data from a high-resolution (1/12°) ocean sea-ice hindcast[23] and employed an Eulerian surface water-mass transformation framework[24-26] to 1) Evaluate the contributions of Atlantic inflows and Arctic outflows to the AMOC lower limb; 2) Identify and quantify the role of air-sea fluxes and interior mixing in transforming the Atlantic and Arctic waters.

## Results

### Pathways and along-path water mass properties

The total southward volume transport across Fram Strait (Fig. 1a, red line) is 8.6 Sv (Table 1). This estimate matches well with previous observational and modelling studies[27,28]. A majority of the southward transport, i.e., 5.6 Sv (65%, Table 1), reaches the eastern SPNA section (Fig. 1a, blue line). The northward volume transport across the eastern SPNA section (Fig. 1b, blue line) is 65.4 Sv in the present study (Table 1), out of which 58.8 Sv (90%) returned to the starting section. These quantitative results compare remarkably well with a recent study[16]. The amount of volume transport reaching the Barents Sea (Fig. 1a, orange line); the sea surface (evaporating); and Fram Strait from these two sets of Lagrangian experiments is provided in Table 1.

The pathways associated with the Arctic outflow and Atlantic inflow are obtained from the Lagrangian trajectories and are mapped in Fig. 2a, b. The spread of Arctic waters in the Nordic Seas and eastern SPNA (Fig. 2a) is associated with a diverse set of pathways: East Greenland Current (EGC)[29]; Jan Mayen Current (JMC)[30]; Greenland Sea gyre[31]; East Icelandic Current (EIC)[32]; shelf break EGC[33]; separated EGC[34]. The Arctic waters along the Jan Mayen Ridge (JMR) and EIC flows together in Iceland-Faroe Slope Jet (IFSJ)[18] and pass through the Faroe-

Shetland Channel (FSC) and Faroe Bank Channel (FBC) to reach the eastern SPNA section (Fig. 2a). The Norwegian Atlantic Front Current (NwAFC) and Norwegian Atlantic Slope Current (NwASC)[35,36] are noted to be transporting the Arctic waters back towards the Fram Strait. The Atlantic inflow primarily moves with the eastern subpolar gyre circulation (Fig. 2b). Norwegian Atlantic Slope Current and North Icelandic Irminger Current (NIIC) are found to be responsible in transporting Atlantic waters into the Nordic Seas (Fig. 2b).

The watermass properties of the Arctic outflow (for Atlantic inflow, see Supplementary Fig. 1) reveal that along the east Greenland coast Arctic waters are low in temperature, fresh, and less dense (Fig. 2c–e), indicating the control of salinity on the density field. However, as the Arctic waters reach the eastern part of the Nordic Seas and the eastern SPNA, they get characterized by deeper, warmer, saltier, and denser watermass properties than found in Fram Strait, associated with a larger mixed layer depth (Fig. 2c–g). Waters exiting Fram Strait are generally colder, fresher and thus must gain heat and salt to become so much warmer and saltier water masses. The larger depth associated with these locations (Fig. 2f) indicates that the gain of heat and salt is due to mixing with deeper, warmer, and saltier non-arctic watermass, carried into the region by the North Atlantic Current (NAC), the Norwegian Atlantic Slope Current and the Norwegian Atlantic Front Current. The age of Arctic water masses increases as they move away from the starting section i.e. Fram Strait (Fig. 2h). Note that the shelf break EGC takes less than 1000 days to reach the eastern SPNA section, while the travel time is much longer for the separated EGC (Fig. 2h). This disparity of timescales associated with the two branches is related to the depth of the flow (Fig. 2f, deeper depths correspond to slower circulation).

### Interaction between the Arctic outflow and the Atlantic inflow

To focus on dense waters that are leaving the eastern SPNA as the AMOC lower limb, the meridional overturning circulation (MOC) has been computed with trajectories that had reached the eastern SPNA section (Fig. 3). It is clearly evident that the majority of Atlantic waters enter the eastern SPNA as warmer, saltier and lighter water masses (Fig. 3a–c, orange contours). After entering, these waters first undergo gradual densification (Fig. 3c) due to the atmospheric cooling (Fig. 3a), while freshening as a result of the net freshwater input (Fig. 3b). The modified waters subsequently sink to greater depths, with a fraction recirculating southward. The remaining portion of the densified water continues northward until it comes in contact with the relative colder and fresher Arctic outflow present in Denmark Strait (between ≈ 65 °N to 68 °N). Around Denmark Strait, the Atlantic inflow is losing heat and salt, and further increasing density, while Arctic outflow is subject to opposite tendencies (Fig. 3a–c). So, the sub-surface mixing of Atlantic waters is necessary for the AMOC lower limb to achieve its density and

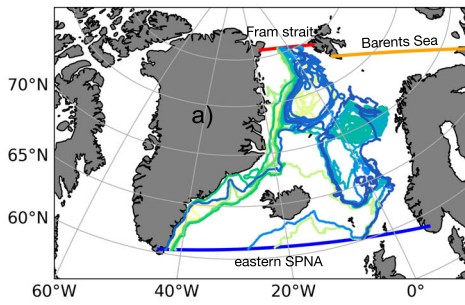
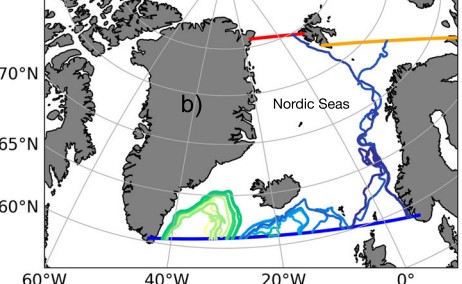
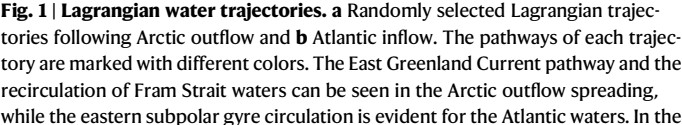

**Fig. 1 | Lagrangian water trajectories. a** Randomly selected Lagrangian trajectories following Arctic outflow and **b** Atlantic inflow. The pathways of each trajectory are marked with different colors. The East Greenland Current pathway and the recirculation of Fram Strait waters can be seen in the Arctic outflow spreading, while the eastern subpolar gyre circulation is evident for the Atlantic waters. In the present work, southward transports are seeded across Fram Strait (red line) and northward volume transports are initiated along the eastern Subpolar North Atlantic Ocean (SPNA) section (blue line). The ending sections of the trajectories are indicated by red, blue and orange lines.

**Table 1 | The total volume transport quantification (unit is 1 Sv ≡ 10⁶ m³ s⁻¹) of Arctic outflow and Atlantic inflow reaching the eastern Subpolar North Atlantic Ocean (SPNA), Fram strait, and Barents Sea, both at the surface (evaporated) and still circulating**

| Starting Sections | Total | eastern SPNA | Fram strait | Barents Sea | Evaporated | Still circulating |
|---|---|---|---|---|---|---|
| Fram strait (Arctic outflow) | 8.6 Sv | 5.6 Sv (65%) | 2.3 Sv | 0.06 Sv | 0.02 Sv | 0.62 Sv |
| eastern SPNA (Atlantic inflow) | 65.4 Sv | 58.8 Sv (90%) | 3.1 Sv | 3.5 Sv | 0 Sv | 0Sv |

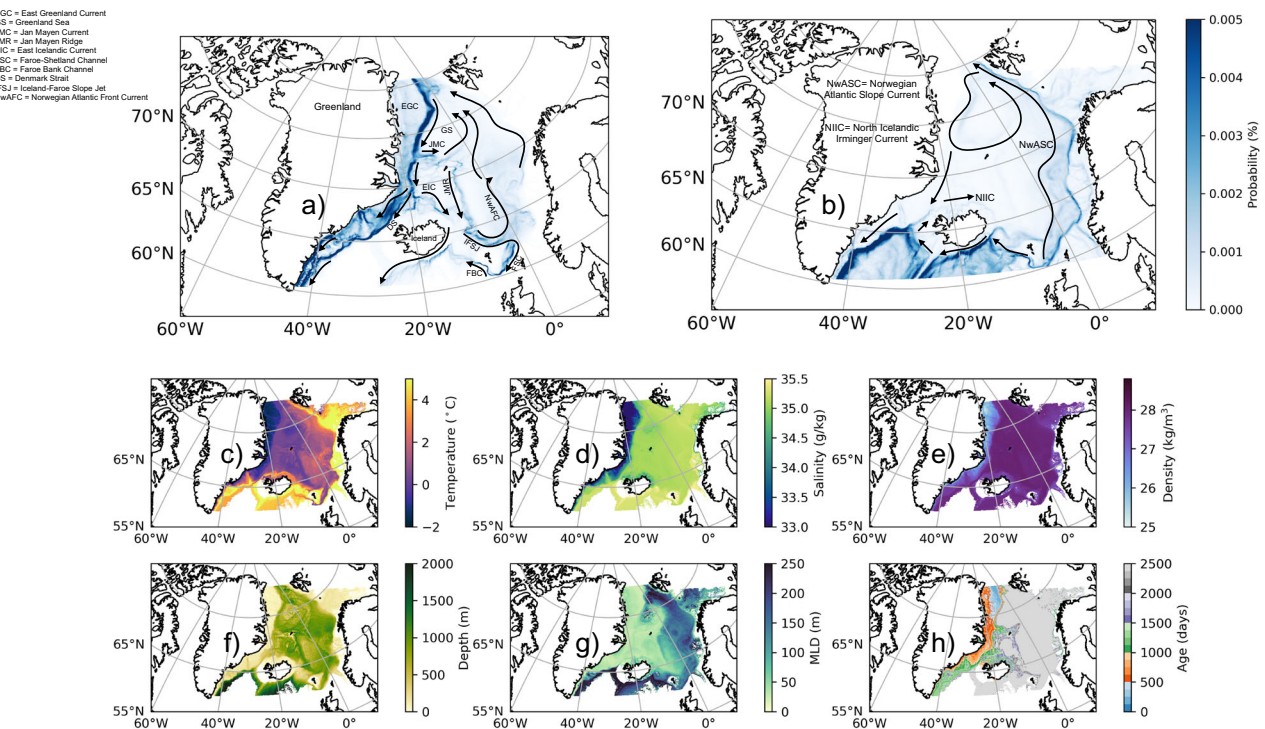

**Fig. 2 | Lagrangian water transport pathways and the value of tracers along with it. a** Parcel position probability map computed (using equation (1)) from all the southward flowing Arctic waters through Fram Strait. This shows the dominant pathways taken by the Arctic waters in the eastern Subpolar North Atlantic Ocean (SPNA) and Nordic Seas. Here EGC = East Greenland Current; GS = Greenland Sea; JMC = Jan Mayen Current; JMR = Jan Mayen Ridge; EIC = East Icelandic Current; FSC = Faroe-Shetland Channel; FBC = Faroe Bank Channel; DS = Denmark Strait; IFSJ = Iceland-Faroe Slope Jet; NwAFC = Norwegian Atlantic Front Current. **b** Same as previous but for the northward flowing Atlantic inflows through the eastern SPNA section. Here NwASC = Norwegian Atlantic Slope Current; NIIC = North Icelandic Irminger Current. Along-path water-mass properties obtained from Lagrangian trajectories that correspond to Arctic outflow. All trajectories were considered that have reached one of the three ending zones (as shown by red, blue and orange lines in Fig. 1). All depth levels, associated with trajectories, are considered in this computation.The panels corresponds to **c** Conservative temperature (°C), **d** absolute salinity (g kg⁻¹), **e** potential density referenced to the surface (kg m⁻³), **f** Depth (m), **g** Mixed layer depth (MLD, in m), **h** Age (days; calculated using equation (2)).

depth. At low temperatures, the density of these Arctic waters is primarily determined by salinity (Fig. 3c, blue dashed contours). On encountering relatively warmer and saltier Atlantic waters between ≈ 60 °N to 63 °N, Arctic waters continue to gain heat and salt (Fig. 3a–c). These latitudes coinide with those of the Faroe Bank channel and FSC.

The changing properties of the two water masses with an opposite sign along the Faroe Bank channel, FSC and the Denmark strait confirms the exchange of heat and salt between them. Finally, after mixing, the Atlantic and Arctic water masses exit the eastern SPNA with similar temperature, salinity, and density as the lower limb of the AMOC (Fig. 3a–c). Further analysis of the overturning stream function in the density-latitude coordinate (Fig. 3c, orange contours) reveals that the maximum of the overturning stream function at the eastern SPNA section (southernmost latitude of Fig. 3c) is 10.1 Sv on the 27.57 kg m⁻³ isopycnal for the North Atlantic inflow. This means that 10.1 Sv of Atlantic waters has transformed from lower densities (<27.57 kg m⁻³ isopycnal) to densities higher than 27.57 kg m⁻³ (lower limb of the AMOC). The low-to-high density watermass transformation of the Atlantic waters is a combined effect of air-sea fluxes and interior

mixing with Arctic waters. A small overturning cell is also present in density space with an amplitude of 0.7 Sv in which Atlantic waters with densities higher than 27.89 kg m⁻³ overturn, becoming waters with densities between 27.89 kg m⁻³ and 27.57 kg m⁻³ (Fig. 3c, anticlockwise rotating cell) due to mixing.

In summary, the total transformation rate of Atlantic waters into the lower limb of AMOC is 10.8 Sv on the 27.57 kg m⁻³ isopycnal. The amount of Arctic water that reaches the eastern SPNA transect with densities >27.57 kg m⁻³ and contributes to the lower branch of the AMOC is 4.3 Sv. Thus, the total time-mean AMOC in the eSPNA is 15.1 Sv. These numbers are in agreement with previous observational evidence[8,14,15]. The density-latitude MOC obtained from tracing the Atlantic waters does not involve a dominant watermass transformation from low-to-high density along the eastern periphery of the Nordic Seas (i.e. north of 70 °N in Fig. 3c) and this branch is therefore considered to be a minor component of the time-mean AMOC.

The meridional overturning stream function in density-latitude coordinates (Fig. 3c) shows that the AMOC lower limb formation is mostly located between ≈ 60 °N and 70 °N, and that Arctic waters

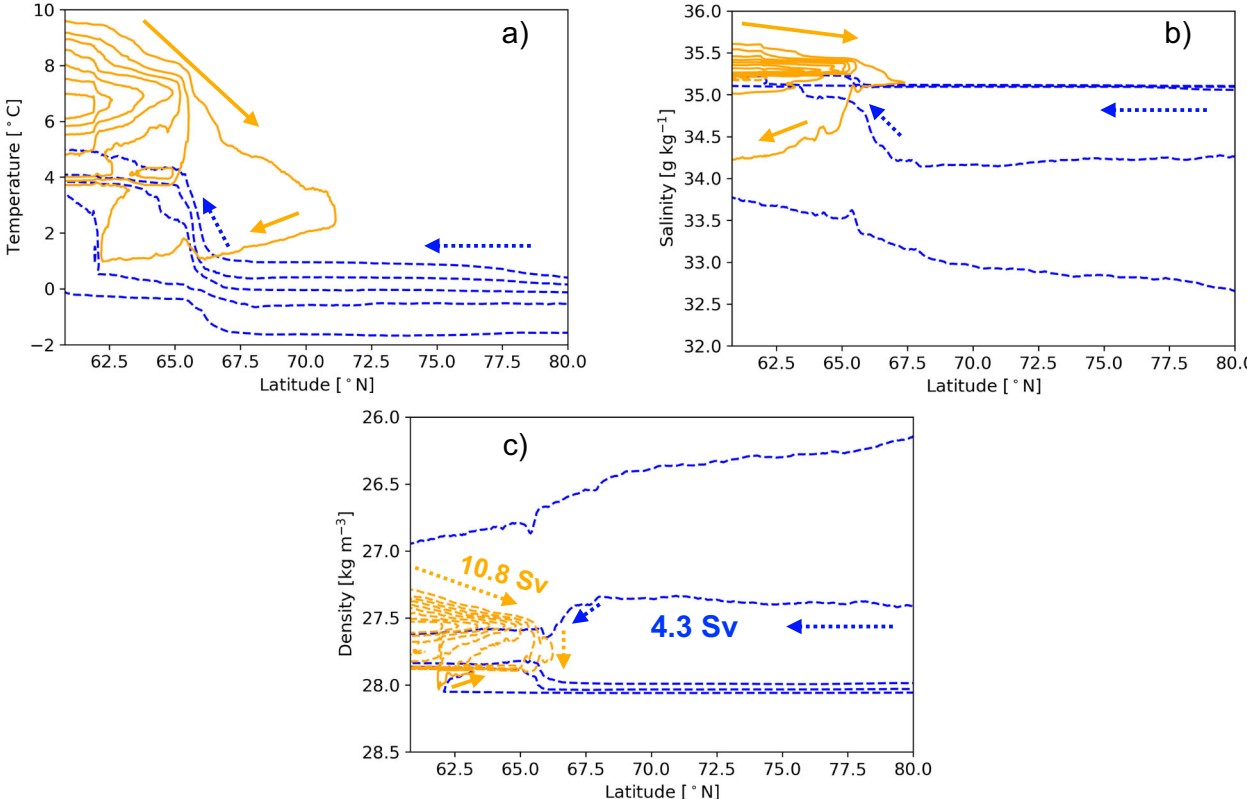

**Fig. 3 | Lagrangian meridional overturning stream function. a** Meridional overturning stream function in a) temperature-latitude, **b** salinity-latitude, and **c** density-latitude coordinate system. This has been computed using equation (3) and with trajectories that have exited the eastern Subpolar North Atlantic Ocean (SPNA) section southward. The orange lines indicate Atlantic inflows and blue dashed lines correspond to Arctic waters that have reached the eastern SPNA. The direction of the water movement in each figure is indicated by arrows. The solid streamlines are 2 Sv apart and start at 1 Sv. Similarly, the dashed streamlines are plotted with 1 Sv interval and start at -1 Sv. The Atlantic and Arctic water contribution to the Atlantic Meridional Overturning Circulation (AMOC) lower limb are shown with bold orange and blue numbers respectively.

contribute to the AMOC lower limb through changes in temperature, salinity, and density from Denmark Strait onwards, where they encounter Atlantic inflows. Interior mixing is found to be the dominant process in transforming Arctic waters that join the AMOC lower limb, while Atlantic waters are transformed through a combination of air-sea fluxes and interior mixing. The Eulerian surface watermass transformation framework (calculated using equation (10) and between ≈ 60 °N and 70 °N) revealed that 7.2 Sv of Atlantic waters (52% of total AMOC and 67% of Atlantic waters overturning) are transformed to lower limb (densities >27.57 kg m⁻³), due to air-sea interactions. This implies the remaining 3.6 Sv (≈ 33%) of Atlantic waters must have been transformed through interior mixing with Arctic waters. The Lagrangian heat and salt divergences in Fig. 4 and in Supplementary Fig. 2, 3 confirm the heat and salt exchanges between the Arctic waters (gaining) and Atlantic waters (losing) between ≈ 60 °N and 70 °N. It also affirms that the Arctic water transformation is only due to the interior mixing as it gains heat and salt primarily below the mixed layer depth (MLD) (Fig. 4, top panel and Supplementary Fig. 2) where Atlantic inflows lose heat and salt (Fig. 4, bottom panel and Supplementary Fig. 3). Here (Fig. 4, bottom panel and Supplementary Fig. 3) also one can see that closer to the surface (within the MLD) the Atlantic inflows are not significantly losing heat and salt along the eastern periphery of Nordic Seas (i.e. north of 70 °N). It is worth noting that Atlantic waters are also losing heat below the MLD downstream of Denmark Strait (Supplementary Fig. 3). However, the Arctic waters are not gaining heat at these locations (Supplementary Fig. 2). This indicates that the heat loss of Atlantic waters at these locations is occurring just below the base of the MLD and is due to the air-sea fluxes. For an example, let us

consider a winter-time trajectory within the MLD enters a grid box through its western grid wall, moves just below the MLD, and exits through the eastern wall. The trajectory will thus experience the wintertime cooling before it leaves the grid box. When calculating the heat divergence at the exit wall (or in other words below the MLD), a cooling of the parcel will be observed. Thus, although the total heat divergence obtained from the Atlantic inflows is mostly determined by the below MLD heat divergence (Fig. 4, bottom panel), it should be considered as a result of mixing if the Arctic waters are gaining heat at the same locations.

## Dependency on chosen time period
The Atlantic and Arctic Ocean contributions to the AMOC lower limb can depend on the chosen time period, as individual Lagrangian water parcels are expected to reach the eastern SPNA section at different times due to the unique ocean currents, hence air-sea fluxes and mixing. Consider two water parcels moving northward from the eSPNA section with contrasting flows. One parcel moves with strong near-surface ocean currents and avoids surface densification during winter, to subsequently exit the eastern SPNA section southward. In contrast, the other parcel moves slowly and is surface-densified during winter before leaving the eSPNA. In the former case, the water parcel will not contribute to the AMOC lower limb as it cannot acquire sufficiently high density, while the latter water parcel is fully transformed to the density of AMOC lower limb water. As the seasonal cycle also varies from year to year, we anticipate changing Atlantic and Arctic contributions to the AMOC lower limb, depending on the time period under consideration. We can further examine whether forward and

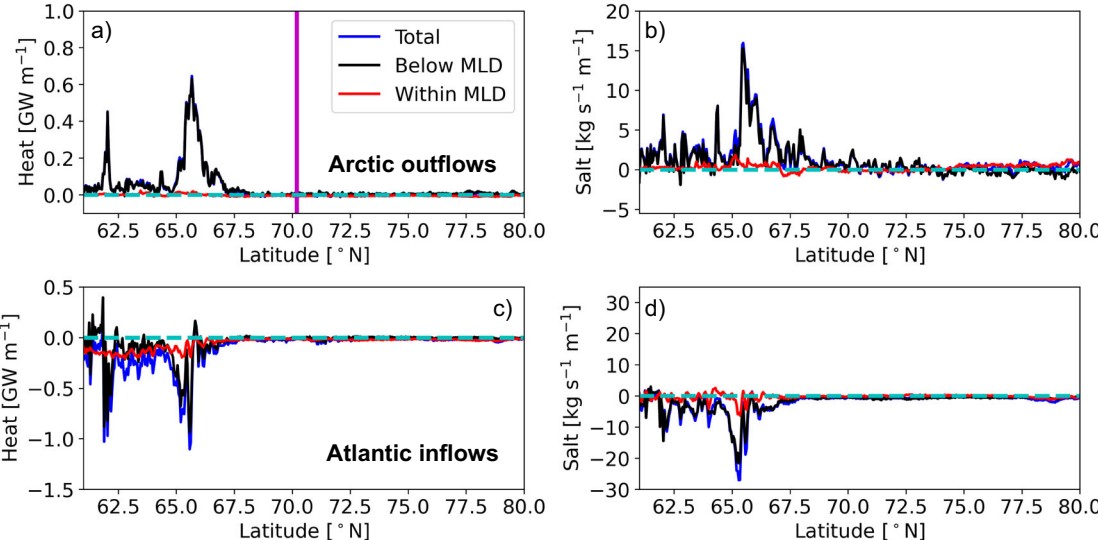

**Fig. 4 | The Lagrangian heat and salt divergence.** Zonally integrated Lagrangian heat (GW m⁻¹; using equation (5)) and salt (kg s⁻¹ m⁻¹; derived from equation (7)) divergence of the Arctic outflows (**a**, **b**) and Atlantic inflows (**c**, **d**) that have exited the eastern Subpolar North Atlantic Ocean (SPNA) southward. The blue, black, and red line corresponds to total divergence, divergence below the mixed layer depth (MLD), and divergence within the MLD respectively. The magenta line shows the northern latitudinal limit ( ≈ 70 °N) within which we have employed the Eulerian surface watermass transformation framework. The cyan color indicates no divergence in heat and salt.

backward tracking of ocean water parcels provides similar results. To address the time dependency of our key findings, we therefore back-track waters flowing southward across the eSPNA section in 2021, for comparison with forward tracking from 1979.

For the backward tracking experiment, we obtain a counterpart meridional overturning stream function in property-latitude coordinates along with contributions of Atlantic and Arctic waters to the AMOC lower limb (Supplementary Fig. 4) that closely resemble those obtained for the forward tracking experiment (Fig. 3). This strengthens our conclusions and demonstrates the robustness of the results. A detailed inspection of the 2021 back-tracking experiment revals the waters from the subtropical areas are warmer and lighter (Supplementary Fig. 4a, c) than in the 1979 forward-tracking experiment. This may be associated with the time that water parcels spend in the eastern SPNA. In the forward tracking experiment, more than 90% of northward-moving waters across the eSPNA took around 5 years to exit southward (i.e. by 1979 + 5 = 1984, Supplementary Fig. 5). The time-scale for the majority of backward-tracked parcels to arrive at the eSPNA section is also around 5 years, but that will be by 2016 (2021-5). We conclude that warmer water prevails around the eSPNA section later in the hindcast. Small differences in the stream functions obtained from forward and backward tracking may be more generally explained by variability of air-sea fluxes and interior mixing throughout the hindcast. However, only the Atlantic contribution to the AMOC lower limb is notably different between forward and backward tracing experiments, the Arctic contribution being steady. Backward tracing reveals that the Atlantic and Arctic contribution to the AMOC lower limb have uncertainty of 3% as compared to the forward trajectory simulation. This difference is attributed to the exposure of Atlantic waters to variable air-sea interaction, while the Arctic waters are only subject to steadier interior mixing. Backward tracking of southward flows across the eSPNA is furthermore insufficient to fully quantify combined transport from the Fram Strait and eSPNA sections (and respective partitioning), compared to the forward trajectories. Based on our explanation, it is important to note that the estimate of 67% of Atlantic waters overturning due to air-sea fluxes and 33% related to interior mixing is accurate to within a few percent, clearly highlighting the relative importance of air-sea interactions and mixing processes.

The noted differences in the stream functions obtained from forward and backward Lagrangian tracking might also be associated with the inability to start trajectories from their exact ending position after a forward/backward tracking simulation using time-evolving fields. This is since we are limited by data unavailability at interpolated time steps in the model/reanalysis. Consider for example a trajectory that starts at the Fram Strait in January 1979 (using monthly data from the hindcast model) and reaches the eSPNA section after 11.3 years. Although we know the ending position of that trajectory, we cannot start a backward run from the same location, as the hindcast model (or any model/reanalysis) does not provide data at an interpolated time step (in the above example, we would need data on day 18.25 of March 1990). The choice of 43 years for trajectory simulations was done to ensure that most trajectories (or volume transport) have had time to exit the study domain (Supplementary Fig. 5). When it comes to the small difference of the stream functions in Fig. 3 and Supplementary Fig. 4 due to mismatch in forward and backward trajectory pathways, it is important to note that the trajectories will not disperse evenly in space but be concentrated where the strong currents are located e.g., the East Greenland Current, Norwegian Atlantic Slope Current, etc (Fig. 2a, b) The similar shape of stream functions obtained from the forward and backward trajectory (Fig. 3 and Supplementary Fig. 4) experiments also validate this argument that trajectories mainly follow the strong mean currents.

## Discussion

Lagrangian analysis of an ocean model hindcast spanning 1979-2021 reveals that the AMOC lower limb consists of 72% Atlantic waters and 28% Arctic waters. The warm and salty Atlantic waters initially become dense due to heat loss to the atmosphere sinking to greater depths. A portion of the deep, dense water returns immediately southward and exits the eastern SPNA. The remaining dense water moves northward and encounters cold and fresh Arctic waters in the Denmark Strait and along the Faroe-Shetland and Faroe Bank channels. There, the Atlantic waters further gain density, sink, and return southward to leave the eastern SPNA.

We find that 67% of the Atlantic water transformation is contributed by the air-sea fluxes, while 33% of overturning is related to the

interior mixing with Arctic waters. We have also shown that low-to-high density water mass transformation of Atlantic waters (that are not entering the Arctic Ocean) along the eastern rim of the Nordic Seas is only a minor component of the AMOC. Previous studies focused on the heat-loss-driven watermass transformation of Atlantic waters around the perimeter of Nordic Seas[15,37,38] and open ocean convection in the Greenland Sea and Iceland Seas[13] but with little or no emphasis on its role in determining the AMOC strength. We show that neither open-ocean convection in the Greenland and Iceland Seas, mixing between gyre and boundary, nor heat loss of ocean waters along the eastern periphery of Nordic Seas (or in other words along the NwAFC and Norwegian Atlantic Slope Current) which do not enter the Arctic Ocean are sufficient to transform waters to the high densities necessary to sustain the AMOC lower limb. However, ocean heat loss along the Nordic Seas periphery can have an indirect effect on the Arctic outflow. After losing heat to the atmosphere, a substantial fraction of densified Atlantic waters travel further northward and enter the Arctic Ocean through Fram Strait ($\approx 3$ Sv) and the Barents Sea ($\approx 3.3$ Sv) as shown in Supplementary Fig. 6, 7. In the Arctic, these Atlantic waters may be further modified through interaction with the atmosphere and Arctic freshwater before being exported southward again through Fram Strait. Such a scenario would be in line with previous findings[10,39]. An extension of the present study could thus be to trace Atlantic waters through the Arctic Ocean, on a longer timescale. This will help to identify and quantify the water mass transformation of Atlantic waters in the Arctic, of indirect consequence for the AMOC.

Given the importance of interior mixing in sustaining a realistic-strength AMOC in the current hindcast, we note that some HighResMIP (High-Resolution Model Intercomparison Project) climate models underestimating AMOC strength are characterised by a shallow lower limb[40]. This highlights the importance of correctly representing both water mass properties (densities), pathways, and transformation processes (surface and interior) in climate models. Global climate model simulations robustly indicate a slowdown of the AMOC in warmer future climate[41–43]. These simulated slowdowns are associated with a combination of surface warming and freshening that reduce surface density[42], increasing upper ocean stratification. We have emphasised here the importance of mixing between Atlantic and Arctic waters, as a substantial component of water mass transformation, necessary to sustain the lower limb and to achieve the density and depth of the return flow. By inhibiting mixing through raised potential energy demand, stronger stratification will reduce the amount of turbulent kinetic energy available for this mixing. Reduced sub-surface mixing of Atlantic waters is thus potentially the key to overall AMOC slowdown, further reducing the depth to which surface overturned Atlantic waters are returned southward. In summary, in a future weakened AMOC, the lower limb will likely shoal to shallower depths, as was apparent during the Last Glacial Maximum[5]. Beyond the direct climate impacts of a weakened AMOC on poleward heat transport, a shoaled lower limb will further shorten timescales for carbon dioxide outgassing from the ocean to the atmosphere.

## Methods

### Hindcast model
To study the southward Arctic outflow through Fram Strait and the northward Atlantic inflow into the eastern SPNA and Nordic Seas, we use the monthly mean model output from the GO8p7-eORCA12 global ocean sea-ice hindcast simulation[23]. The version 8 of the UK Global Ocean (GO) configuration is based on the Nucleus for European Modelling of the Ocean (NEMO) 4.0.4 version ocean model[44]. The extended version of the ORCA12 grid (eORCA12) has an eddy-rich resolution of 1/12° and includes oceanic circulation under the ice cavities in Antarctica. The model has 75 vertical levels and a tri-polar grid with poles located at Canada, Siberia, and Antarctica. The vertical mixing of tracers in the model is parameterised using an improved

version of the Turbulent Kinetic Energy (TKE) scheme[44,45]. The NEMO model uses a nonlinear free surface in which the volume of the ocean grid cells at every vertical level is allowed to vary with time[46]. The sea-ice model used in this configuration is CICE[47] which includes the effect of surface melt ponds. The hindcast simulation was initialised on January 1st, 1958, with 1995–2014 mean temperature and salinity profiles of the EN4 climatology[48], and forced with the Japanese 55-year atmospheric reanalysis (JRA55-do) dataset[49]. The model was integrated from 1958 to 2021 with a model time step of 300s.

### Lagrangian water tracing algorithm TRACMASS
The Lagrangian trajectory code TRACMASS v7.0[50] is used in the present study. Trajectories were calculated off-line, i.e., after the hindcast model saved the thermodynamic and dynamic fields. This allows us to simulate many more trajectories than is possible in an on-line integration. The first working version of TRACMASS was introduced in the 1990s[51] and since then it has been updated and adapted regularly. TRACMASS advects trajectory using volume/mass transport fields instead of velocity fields and is thus able to conserve mass/volume. This makes it possible to compute Lagrangian stream functions and Lagrangian divergence[52]. For time-dependent mass/volume transport fields, TRACMASS simulates trajectory routes with an analytical solution by solving a differential equation that is based on linear interpolation of mass/volume transports on the model grid box walls[53]. This scheme is known as stepwise-stationary, in which mass/volume transport fields are assumed to be stationary during intermediate time steps between two model output sets[54]. TRACMASS computes vertical mass/volume transports from the continuity equation and is thus a mass-conserving algorithm. This characteristic helps the simulated trajectories to stay away from solid boundaries such as the coast or ocean bottom. A trajectory calculated by TRACMASS can be viewed as a streamtube. In the absence of stochastic parameterisation, the volume transport at one end of the streamtube must be equal to the volume transport at the other end (non-divergent)[55]. Thus, the streamtubes are equivalent to the volume transport pathways of an incompressible fluid. Sub-grid scale processes are not parameterized during the TRACMASS analyses. Parcel properties (temperature, salinity) do however evolve along trajectories, as a consequence of parameterized mixing and air-sea interaction in the hindcast model. TRACMASS has been used in numerous studies to understand the different branches of the large-scale ocean circulation and the associated water mass transformation[16,52,56–59].

### TRACMASS configuration used in the study
Two different trajectory runs were performed using TRACMASS to identify the water transport routes and to quantify the water mass transformation associated with the Arctic outflow and Atlantic inflow into the eastern SPNA and Nordic seas (Fig. 1a–b). The first simulation deals with trajectories starting in southward direction from Fram Strait (Fig. 1a; Arctic outflow); the second deals with trajectories starting in northward direction from the eastern SPNA section (Fig. 1b; Atlantic inflow). In both simulations trajectories were seeded across all 12 months of the year 1979 and advected forward in time for 43 years, i.e., until 2021 to represent the present-day climate. Note that the original hindcast model data is available from 1958 to 2021 but we discarded the first 21 years to minimize the effect of model drift on the results. Additionally, the JRA55-do atmospheric forcing dataset uses satellite products for bias-correction and thus more realistic 1979 onward[49]. In the Arctic outflow run, southward flowing water parcels were initiated across Fram Strait (Fig. 1a, red line) through all depths. A volume transport of maximal 1500 m³s⁻¹ was assigned to each trajectory. This implies that when the southward volume transport in a grid box exceeds 1500 m³s⁻¹, the number of water parcels seeded was equal to the volume transport divided by 1500, rounded off upwards to an integer value. Using 1500 m³s⁻¹ as target resolution per water parcel

ensures a good Lagrangian resolution[55]. Similarly, during the Atlantic inflow simulation, northward flowing water parcels were initiated across the whole depth range (Fig. 1b, blue line) with each parcel carrying a volume transport less than 2500 $m^3 s^{-1}$. The larger number here is associated with the much larger total transport associated with the Atlantic inflow, keeping the resolution of this flow still very high. A sensitivity test was performed through varying the maximum volume transport by ± 500 $m^3 s^{-1}$ for the two Lagrangian experiments. The results remained unchanged within 1% uncertainty levels. Since NEMO uses a quasi-isotropic grid, the latitude is not fixed along a constant meridional model index. In each of the Lagrangian runs, trajectories were halted when they reached Fram Strait (Fig. 1a-b, red line) or reached the eastern SPNA section (Fig. 1a–b, blue line), or entered the Barents Sea (Fig. 1a–b, orange line). Trajectories exiting through the sea surface (i.e. evaporating) and the trajectories still circulating without reaching any of the ending sections were discarded. A total of 0.02 Sv evaporated and 0.62 Sv is still circulating in the Arctic outflow simulation (Table 1). Conservative temperature (°C), absolute salinity (g $kg^{-1}$), potential density (kg $m^{-3}$), and mixed layer depth (m) were stored along the trajectory pathways and at the sidewalls during each grid cell crossing. The conservative temperature, absolute salinity, and mixed layer depth were taken directly from the model monthly mean outputs. The potential density referenced to the surface was computed by using the TEOS-10 equation of state[60,61], as in the GO8p7-eORCA12 configuration. The mixed layer depth was computed in the NEMO model using a density criterion[62].

### Lagrangian pathways, stream function, and heat and salt flux divergence

To each trajectory simulated by TRACMASS a volume transport was assigned, which is conserved throughout its journey due to the volume-conserving algorithm. This makes it possible to compute a stream function from Lagrangian trajectories, as there will be no sources and sinks of water between start and end[52,56]. The water circulation pathways in the longitude-latitude framework were identified by constructing a parcel position probability ($P_{i,j}$) map[55]:

$$P_{i,j} = \frac{P_{i,j}^c}{\sum P^c} \times 100\%, \qquad (1)$$

where $P_{i,j}^c$ is the parcel count at a particular longitude ($i$), latitude ($j$) grid box, and $P^c$ denotes a total number of parcel counts for all the grid boxes over the whole integration period. The unit of $P_{i,j}$ is in % per horizontal grid box. The sum over all $P_{i,j}$ is thus 100%. Once ocean water parcels start to move away from their starting location, an age ($\lambda_{i,j}$) along their pathways has been recorded:

$$\lambda_{i,j} = \frac{\sum_{m=1}^{M}\left\{\left(t_{i,j}^m - t_S^m\right) \cdot F_m\right\}}{\sum_{m=1}^{M} F_m}, \qquad (2)$$

where $t^m$ is time when the $m^{th}$ trajectory crosses a grid-box wall with a mass transport of $F_m$ and $t_s^m$ is the time at the start. The meridional overturning stream function in tracer-latitude space was computed using the following equation:

$$\psi_{j,r} = \sum_{r'=rmax}^{rmin} \sum_i \sum_m F_{i,j,r',m}^y. \qquad (3)$$

Here $F_{i,j,r',m}^y$ is the volume transport at a particular longitude ($i$), latitude ($j$), and tracer bin ($r'$), carried by the trajectory with index $m$, crossing latitude $y$. The volume transport of the trajectories was integrated zonally and assigned to a certain tracer bin by integrated it from the maximum tracer value to the minimum value for each tracer bin. In this way, we obtain volume transport as a function of tracer value in the

same way as e.g., an overturning stream function as a function of potential density is estimated in Eulerian calculations. Here, the tracers we use are conservative temperature, absolute salinity, and potential density. The stream functions in the tracer-latitude framework are useful to visualise and quantify the water mass overturning rate in the ocean[56,58].

To identify the locations where oceanic waters are gaining/losing heat flux, the Lagrangian divergence of advective heat flux ($H_{i,j}^L$) was computed from the simulated trajectories[52]

$$H_{i,j}^L = \sum_k \sum_m F_{i,j,k,m}\left(T_{i,j,k,m}^{out} - T_{i,j,k,m}^{in}\right) \times \frac{\rho c_p}{\Delta x_{i,j}\Delta y_{i,j}}, \qquad (4)$$

where $F_{i,j,k,m}$ is the volume transport through any of the six grid box walls by a trajectory with index $m$. $T_{i,j,k,m}^{out}$ and $T_{i,j,k,m}^{in}$ are the conservative temperature a trajectory carries as it leaves and enters a grid box, respectively. A positive value of $H_{i,j}^L$ will thus indicate heat-gain and the opposite holds true for negative values. A change in advective heat flux can either be due to air-sea interaction or by exchanging heat with other water masses, e.g., mixing. In equation (4), the density of the seawater was taken as 1026 kg $m^{-3}$ and the specific heat capacity as 3992 J $kg^{-1}$ $C^{-1}$. $\Delta x$ and $\Delta y$ represent the zonal and meridional grid spacing. The latitudinal distribution of the advective heat flux divergence is obtained by zonal integration[63]:

$$H_j^M = \sum_i \left(H_{i,j}^L \times \Delta x_{i,j}\right). \qquad (5)$$

The Lagrangian advective heat divergence $H_{i,j}^L$ has a unit of W $m^{-2}$ and its zonal integrated value $H_j^M$ is expressed in W $m^{-1}$. Similarly, the Lagrangian salt divergence ($S_{i,j}^L$) was calculated as follows

$$S_{i,j}^L = \sum_k \sum_m F_{i,j,k,m}\left(S_{i,j,k,m}^{out} - S_{i,j,k,m}^{in}\right) \times \frac{\rho}{\Delta x_{i,j}\Delta y_{i,j}}, \qquad (6)$$

where $S$ denotes absolute salinity and $S_{i,j}^L$ has a unit in g $s^{-1}$ $m^{-2}$. Changes in the advective salt flux can be due to the evaporation, precipitation, freezing of ice, melting of ice, river runoff, or through salt exchange with other water masses (mixing). The latitudinal distribution of the salt flux divergence has been calculated as:

$$S_j^M = \sum_i \left(S_{i,j}^L \times \Delta x_{i,j}\right), \qquad (7)$$

where $S_j^M$ is described in g $s^{-1}$ $m^{-1}$.

### Eulerian surface water-mass transformation

To quantify the amount of water-mass transformation between ≈ 60 °N and 70 °N due to the net surface heat and freshwater fluxes, a water-mass transformation framework was implemented[24–26,64]. The net surface density flux $F_{surface}^\rho$ is

$$F_{surface}^\rho = -\frac{\alpha}{C_p}F_Q + \beta\frac{(E-P)S}{1-S}, \qquad (8)$$

where $\alpha$ is the thermal expansion coefficient (°$C^{-1}$) and $\beta$ is the dimensionless haline contraction coefficient in (kg $kg^{-1}$)$^{-1}$. The $\alpha$ and $\beta$ were computed from the sea surface conservative temperature and sea surface absolute salinity ($S$) using TEOS-10. Here $F_Q$ is the net surface heat flux (units of W $m^{-2}$, positive into the ocean), and Evaporation ($E$) - Precipitation ($P$) is the net water flux (kg $m^{-2}s^{-1}$) out of sea ice and seawater. All the terms in equation (8) are functions of space and time and $F_{surface}^\rho$ has units in kg $m^{-2}s^{-1}$. The diapycnal density flux ($D_{surface}$) is then obtained by area-integrating the $F_{surface}^\rho$ where surface density

$\sigma_{surface}$ is above a given value of density $\sigma$

$$D_{surface}(\sigma) = \frac{1}{N} \sum_{n=1}^{N} \sum_{i=x_w}^{x_e} \sum_{j=y_s}^{y_n} \left( F^{\rho}_{surface}(i,j,n) \times \Delta x_{i,j} \times \Delta y_{i,j} \right). \qquad (9)$$

Here $x_w$ and $x_e$ are the western and eastern boundaries, $y_s$ and $y_n$ are the southern and northern limits. If the surface density $\sigma_{surface}$ is not above the given density $\sigma$ then $D_{surface}(\sigma)$ is equals to 0. Here $N$ is the total number of ocean data snapshots (monthly data from 1979 to 2021), which is 516. The diapycnal density flux has units of kg s$^{-1}$. Finally the surface water-mass transformation rate ($G_{surface}$) was computed by taking differences between the diapycnal density fluxes across two density surfaces,

$$G_{surface}(\sigma) = \frac{D_{surface}(\sigma) - D_{surface}(\sigma + \Delta\sigma)}{\Delta\sigma}. \qquad (10)$$

Here density $\sigma$ is taken between 23 kg m$^{-3}$ and 29 kg m$^{-3}$ with an increment ($\Delta\sigma$) of 0.01 kg m$^{-3}$.

## Data availability

The raw Lagrangian trajectory data files generated in this study are freely available at https://doi.org/10.5281/zenodo.7924420[65] and https://doi.org/10.5281/zenodo.10210320[66]. The nemo ocean-sea ice hindcast model data can be obtained from https://doi.org/10.5285/399b0f762a004657a411a9ea7203493a[23].

## Code availability

The Lagrangian trajectory code TRACMASS v7.0 is available at https://doi.org/10.5281/zenodo.4337926[50] and also in open source GitHub repository https://github.com/TRACMASS/tracmass[67]. The analysis scripts used to generate the figures have been archived and are available from the contact author DD, d.dey@soton.ac.uk.

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

## Acknowledgements

DD, RM, and SD has been funded by the UK Natural Environmental Research Council (NERC) under Consequences of Arctic Warming for European Climate and Extreme Weather (ArctiCONNECT) Grant no. NE/V005855/1. SJ, BS, and JG acknowledge the ArctiCONNECT grant through agreement no. NE/V004875/1. This work used JASMIN, the UK collaborative data analysis facility. DD would like to acknowledge Dr. Jeffrey Blundell from the University of Southampton, UK for his unconditional support in setting up TRACMASS on JASMIN. The ORCA12 hindcast was led by Dr. Alex Megann of the National Oceanography Centre. DD extends a special thank you to Dr. Jenny Mecking from the National Oceanography Centre, UK for her help in establishing Jupyter Notebook connection between the local computer and JASMIN. Finally, Dr. Adam Blaker from the National Oceanography Centre, UK is acknowledged for providing the data location in JASMIN, and Dr. Aitor Aldama Campino from Swedish Meteorological and Hydrological Institute, Norrköping, Sweden for all the intriguing discussions on TRACMASS.

## Author contributions

D.D. and R.M. together conceived the idea of the present work. S.D. encouraged the use of hindcast model dataset for the current study. D.D. conducted the Lagrangian simulations and carried out analysis. K.D. helped in computing the Lagrangian heat and salt divergences. The results were discussed regularly between D.D., R.M., S.J., B.S., J.G., K.D. and S.D. in order to achieve the study objectives. DD wrote the text with inputs from all the co-authors.

## Competing interests

The authors declare no competing interests.
