## [Peer Review File · Nature Communications]

Formation of the Atlantic Meridional Overturning Circulation lower limb is critically dependent on Atlantic-Arctic mixingREVIEWER COMMENTS

Reviewer #1 (Remarks to the Author):

The paper by Dey et al. investigates the formation of lower limb of north Atlantic Meridional Overturning Circulation (AMOC) using a Lagrangian approach. Two different trajectory runs are performed: the Arctic outflow across Fram Strait and the Atlantic inflow through the eastern Subpolar North Atlantic Ocean (eSPNA). The obtained meridional overturning stream function suggests the role of Atlantic-Arctic mixing south of 70 °N in determining the time-mean AMOC strength. The paper is clearly written, logically presented, and should be a valuable contribution with both regional and larger scale interest. Before possible publication there are some major points that the authors should address.

Major points

1. In addition to the Atlantic-Arctic mixing, I wonder the role of boundary-interior mixing between the Atlantic/Arctic Water (mostly along boundary region) and the Nordic Seas gyre water in contributing to the AMOC strength. Figure 2g shows very shallow mixed layer depth (<150m) in the interior of Greenland and Iceland Seas for the Arctic properties. Are the results of mixed layer depth similar for the Atlantic run? Although the production of dense water in the gyre region of Greenland and Iceland Seas is O (1 Sv), the produced water can very dense (>28.0 kg/m³). The authors need to provide convincing arguments that the densification of Atlantic Water is not due to the mixing between boundary and gyre water in the Nordic Seas.

2. I have concerns about the reproduced Atlantic water pathways in the eastern Nordic Seas in this study. It seems that the Norwegian Atlantic Frontal Current is absent in Figure 2b. However, observational studies have pointed out the significant densification of Atlantic water along this current (Huang et al. 2023) and the important role of overturning loop within the Norwegian Sea (e.g., Eldevik et al. 2009). In addition, previous numerical modeling studies have suggested strong lateral exchange of heat along the Atlantic water pathways in the eastern Nordic Seas, especially for the Lofoten Basin region (68-72° N, e.g., Isachsen et al. 2012 and Spall et al. 2021). This is in contrast with the almost negligible heat divergence of Atlantic inflow north of 68°N (Figure 4). Please justify.

Minor points

1. It would be helpful if the authors could provide general validation of the performance of used hindcast model.

2. Please provide the uncertainty of transport listed in Figure 1c.

3. Why not show the along-pathway water mass properties of Atlantic water run? Perhaps warrant a supplementary figure.

4. Please consider labelling some of streamfunction transport numbers in Figure 3a-c.
5. Why there is a small overturning cell (0.7 Sv) for the dense portion of Atlantic water? Please explain the potential process.
6. Please elaborate the used Eulerian surface water mass transformation framework. In the Eulerian framework, how much of the Atlantic Waters are transformed due to air-sea interactions in the region south/north of 70°N?

References:

Huang, J., Pickart, R.S., Chen, Z. et al. Role of air-sea heat flux on the transformation of Atlantic Water encircling the Nordic Seas. *Nat Commun* 14, 141 (2023).

Eldevik, T. et al. Observed sources and variability of Nordic seas overflow. *Nat. Geosci.* 2, 406–410 (2009).

Isachsen, P. E., Koszalka, I. & LaCasce, J. H. Observed and modeled surface eddy heat fluxes in the eastern Nordic Seas. *J. Geophys. Res. Oceans* 117, C8 (2012).

Spall, M. A., Almansi, M., Huang, J., Haine, T. W. & Pickart, R. S. Lateral redistribution of heat and salt in the Nordic Seas. *Prog. Oceanogr.* 196, 102609 (2021).

Reviewer #2 (Remarks to the Author):

The paper aims to clarify which processes (especially air/sea fluxes and interior mixing) determine the formation of the AMOC lower limb, and how Atlantic and Arctic waters contribute to the AMOC lower limb (Abstract, end of Introduction). These are important, interesting questions in physical oceanography.

There are two main claims: First, the AMOC lower limb critically depends on Atlantic-Arctic mixing (title, Abstract). This is shown in Fig 3, for instance, where the Atlantic inflow cools, freshens, densifies, and flows back south in the SPNA. Second, the Atlantic waters that are transformed in the Nordic Seas are unimportant (Abstract). This is shown in Fig. 3 because little Atlantic water penetrates the Nordic Seas (north of about 66N) and returns south. The paper also quantifies the AMOC lower limb split between Atlantic/Arctic water and the relative importance of air-sea heat fluxes and interior mixing in transforming Atlantic water (page 5).

The qualitative conclusion that mixing in the SPNA is critical to formation of the AMOC lower limb

isn't new. E.g., see the review of Dickson et al. (2008, 10.1007/978-1-4020-6774-7_20) for overflow entrainment to form lower North Atlantic Deep Water, and Straneo (2006, 10.1175/jpo2932.1) for a conceptual model to form upper North Atlantic Deep Water. Nevertheless, the quantitative results from the present study are new, and using a realistic, high-resolution model to track Lagrangian pathways is smart and appropriate. In general the paper's well-written with good organization, clear figures, and sufficient detail.

However, I have concerns about the methods (see below). First, I think the quantitative results from the Lagrangian particles fundamentally depend on the time horizon of the particle trajectories. Choosing a different time horizon will give qualitatively different results. Second, I question the validity of associating a volume transport in Sverdrups to each particle trajectory. This idea underpins the quantitative results in the paper. In short, I think the new results aren't robust.

Hence, I think the paper should be declined by Nature. Comms. in this form. Nevertheless, I encourage the authors to rethink their analyzes to produce quantitative results that are on firmer ground.

Major Comments:

A. The headline result is that the AMOC lower limb consists of "72% of Atlantic and 28% of Arctic waters" (Abstract). However, this result depends on the time horizon of the Lagrangian particle integration. In this case the time horizon is 1979–2021 (43 years, from Page 6 on Methods and Data). The question is: How do the 72%/28% results depend on this time horizon?

In addressing this question, a couple of conceptual issues occur:

The flow is recirculating, so on a long-enough timescale all water comes from everywhere else. Similarly, on a short-enough timescale the AMOC lower limb consists of zero percent Atlantic water and zero percent Arctic water.

Specifically, what the authors call "Arctic" water consists of older "Atlantic" water that's recirculated through the Arctic Ocean. The authors acknowledge this point at the top of page 6. E.g., on a longer time horizon, the orange Atlantic streamlines in Fig. 3 connect to the dashed blue Arctic streamlines coming from 80N. The present particle experiments eliminate this connection from the outset.

Therefore, the 72%/28% split isn't a very meaningful result. A more robust diagnostic of the connectivity from the eSPNA to the AMOC lower limb is to look at the *distribution* of Lagrangian particle time horizons from the Atlantic and Arctic release lines to the AMOC lower limb (and also from the Atlantic release line to the Arctic release line).

B. The computation of overturning stream functions from the Lagrangian particle results using TRACMASS (Fig 3) is interesting, but I question its validity. Specifically:

How does a Lagrangian particle retain its volume transport tag (F_m in equation (2))? This idea is essential to construct the overturning stream functions. For example, on page 7 it says "to each

trajectory simulated by TRACMASS a volume transport was assigned, which is conserved throughout its journey due to the volume conserving algorithm.” There are several unstated assumptions here, which need to be stated explicitly and justified. [SEP] I’m skeptical because I expect nearby particles to diverge exponentially (at least asymptotically, for a while) due to Lagrangian chaotic advection. This is called stirring in the sense of Eckart (1948, J. Mar. Res.); see also LaCasce & Bower (2000, J. Mar. Res.). That means there will be divergence of the ensemble of particles initially tagging the water associated with a particular F_m . Therefore, I don’t understand how a single particle can be associated with a particular F_m .

How is precipitation and evaporation handled? Do particles enter and leave the surface ocean? Or adjust their F_m ?

This issue undermines the stream functions in Fig. 3 and the heat/salt divergences in Fig. 4. In short, I’m worried that the Fig. 3, 4 results aren’t robust.

Minor Comments:

Abstract (first sentence): It says “Deep-water formation in the eastern Subpolar North Atlantic Ocean (eSPNA) and Nordic Seas is crucial for maintaining the lower limb of the Atlantic Meridional Overturning Circulation (AMOC) and stabilizing Earth’s climate.” In what sense is the AMOC “stabilizing” of Earth’s climate? Also, deep water formation occurs in the western SPNA. Why isn’t that mentioned? I suggest this sentence is rewritten and/or better qualified.

Abstract (final sentence): It says “The Atlantic waters that gain density by air-sea fluxes along the eastern periphery of Nordic Seas and do not enter the Arctic Ocean are not critical in determining the time-mean AMOC strength.” This is confusing. Why should these Atlantic waters determine the AMOC “strength” (i.e., transport)? The authors expect something that’s not clear to me here. Also, the penultimate line on page 1, and the last line on page 5 refer to the “AMOC strength” this way.

Introduction (page 1): It says “Dense waters from the Nordic Seas are transported across the Greenland-Scotland Ridge (GSR) to the eSPNA by the East Greenland Current (EGC)”. This is true, but dense waters also cross the GSR east of Iceland, for example, in the Faroe Bank Channel.

Introduction (page 1): Clarify where it says “The other half was found to be associated with oceanic heat loss in the Irminger Sea and Iceland basin.”

Fig. 1: Explain what the colors on the particle trajectories mean. The Barents Sea section is at an unusual angle. Usually, it joins Svalbard to Norway via Bear Island (e.g., Tsubouchi et al., 2021, 10.1038/s41558-020-00941-3).

Fig. 2: What depth is used for the maps in c, d, e, and h?

Fig. 3/page 4: It says that Fig. 3d verifies the 10.8 Sv overturning value. How could Fig. 3d not “verify” the 10.8Sv? Fig. 3d is just a slice of stream function at the left edge of Fig. 3c. It seems to me that it

must, by construction, give a consistent total transformation rate. In other words, I don't think Fig. 4d is an independent verification of the 10.8Sv number.

Fig. 4/page 5: In the text it says that 67% of Atlantic waters are transformed by air-sea interaction and 33% are transformed by interior mixing. But Fig. 4 (bottom row) shows that most Atlantic water transformation occurs beneath the mixed layer (the blue line is very similar to the black line). These results contradict each other. Please explain and reconcile the contradiction.

Page 6: When talking about future AMOC slowdown it says: "Reduced sub-surface mixing of Atlantic waters is a likely part of overall AMOC slowdown". What's the evidence to support this claim?

Page 6, final sentence of main text: It says "The proposed importance of interior mixing can also be helpful to explain why some of the climate models that underestimates the AMOC strength also have shallower AMOC." This needs to be clarified and explained better.

Reply to reviewer 1 *Thank you for the response on our article. We are grateful for all your constructive suggestions, which have helped us improving the manuscript. Below you can see our answers. The line numbers are from the revised manuscript*

Major Comments

In addition to the Atlantic-Arctic mixing, I wonder the role of boundary-interior mixing between the Atlantic/Arctic Water (mostly along boundary region) and the Nordic Seas gyre water in contributing to the AMOC strength. Figure 2g shows very shallow mixed layer depth (<150m) in the interior of Greenland and Iceland Seas for the Arctic properties. Are the results of mixed layer depth similar for the Atlantic run? Although the production of dense water in the gyre region of Greenland and Iceland Seas is O (1 Sv), the produced water can very dense (>28.0 kg/m³). The authors need to provide convincing arguments that the densification of Atlantic Water is not due to the mixing between boundary and gyre water in the Nordic Seas.

Answer : The role of boundary-interior mixing between the Atlantic/Arctic Water (mostly along boundary region) and the Nordic Seas gyre water in contributing to the AMOC strength is found to be minimal in the present study. The meridional overturning stream function in density-latitude framework (Figure 3c) confirms the transformation of Atlantic and Arctic waters to the AMOC lower limb (densities > 27.57 kg m⁻³) between $\approx 60^{\circ}\text{N}$ and 70°N . We have mentioned this on line no: 105 - 106 as “The meridional overturning streamfunction in density-latitude coordinate (Figure 3c) shows that the AMOC lower limb formation is mostly located between $\approx 60^{\circ}\text{N}$ and 70°N ”. Additionally, we have also pointed out on line no: 102 - 104 that “The density-latitude MOC obtained from tracing the Atlantic waters does not involve a dominant watermass transformation from low-to-high density along the eastern periphery of the Nordic Seas (i.e. north of 70°N in Figure 3c) and this branch is therefore considered to be a minor compo-

ment of the time-mean AMOC". Lagrangian heat and salt divergences in Figure 4 and newly added supplementary Fig. 3. also strengthen our argument. We had mentioned on line no: 117 -119 that "Here (Figure 4, bottom panel and Supplementary Fig.3) also one can see that closer to the surface (within the MLD) the Atlantic inflows are not significantly losing heat and salt along the eastern periphery of Nordic Seas (i.e. north of 70 °N)". Furthermore, we have now mentioned explicitly on line no: 140 - 143 that "We show that neither open ocean convection in the Greenland and Iceland Seas, mixing between gyre and boundary in the Nordic Seas, nor heat loss along the eastern periphery of Nordic Seas (or in other words along the NwAFC and Norwegian Atlantic Slope Current) are sufficient to transform waters to the high densities necessary to sustain the AMOC lower limb". Note that in both Lagrangian simulations we did not allow trajectories to enter the Arctic Ocean and thus heat loss along the Nordic Seas periphery can have an indirect effect. We had mentioned this on line no: 143 -148 as "However, ocean heat loss along the Nordic Seas periphery can have an indirect effect on the Arctic outflow. After losing heat to the atmosphere, a substantial fraction of densified Atlantic waters travel further northward and enter the Arctic Ocean through Fram Strait and the Barents Sea. In the Arctic, these Atlantic waters may be further modified through interaction with the atmosphere and Arctic freshwater before being exported southward again through Fram Strait. An extension of the present study could thus be to trace Atlantic waters through the Arctic Ocean. This will help to identify and quantify the water mass transformation of Atlantic waters in the Arctic, of indirect consequence for the AMOC". The mixed layer depth for the Atlantic run is now provided in Supplementary Fig.1 and it is similar to the Arctic simulation in the interior of Greenland Seas. The mixed layer depth is a 43 year time-mean field obtained from the trajectories and thus the wintertime deep mixed layer depth will be somewhat offset by the summertime shallow mixed layer depth.

I have concerns about the reproduced Atlantic water pathways in the eastern Nordic Seas in this study. It seems that the Norwegian Atlantic

Frontal Current is absent in Figure 2b. However, observational studies have pointed out the significant densification of Atlantic water along this current (Huang et al. 2023) and the important role of overturning loop within the Norwegian Sea (e.g., Eldevik et al. 2009). In addition, previous numerical modeling studies have suggested strong lateral exchange of heat along the Atlantic water pathways in the eastern Nordic Seas, especially for the Lofoten Basin region (68-72°N, e.g., Isachsen et al. 2012 and Spall et al. 2021). This is in contrast with the almost negligible heat divergence of Atlantic inflow north of 68°N (Figure 4). Please justify.

Answer : Thank you for raising this point. We indeed have a Norwegian Atlantic Front Current (NwAFC) and is now shown in Figure 2a. The NwAFC is found to be primarily transporting the diverted and modified Arctic waters. Note that from an Eulerian sense one cannot determine if the water in the NwAFC is from the Atlantic or Arctic region. It is important to mention that in both Lagrangian simulations we did not allow trajectories to enter the Arctic Ocean and thus ocean heat loss along the Nordic Seas periphery (or in other words along the NwAFC and Norwegian Atlantic Slope Current) can have an indirect effect. We had mentioned this on line no: 143 - 148 as “However, ocean heat loss along the Nordic Seas periphery can have an indirect effect on the Arctic outflow. After losing heat to the atmosphere, a substantial fraction of densified Atlantic waters travel further northward and enter the Arctic Ocean through Fram Strait and the Barents Sea. In the Arctic, these Atlantic waters may be further modified through interaction with the atmosphere and Arctic freshwater before being exported southward again through Fram Strait. An extension of the present study could thus be to trace Atlantic waters through the Arctic Ocean. This will help to identify and quantify the water mass transformation of Atlantic waters in the Arctic, of indirect consequence for the AMOC”.

Minor points

It would be helpful if the authors could provide general validation of the performance of used hindcast model.

Answer : All GO-designated NEMO ocean model configurations are rigorously developed, tested and validated against a comprehensive set of observational metrics before being used for research and operational use (numerical weather prediction, seasonal and decadal forecasts and historical/future ocean and climate simulations) under the Joint Marine Modelling programme (<https://www.metoffice.gov.uk/research/approach/collaboration/joint-marine-modelling-jmmp%20Model%C2%A0Configurations>), a collaboration between the Met Office, National Oceanography Centre, Centre for Polar Ocean and Modelling and British Antarctic Survey. The general validation of the model configuration is already in preparation (through personal communication with Alex Megann, National Oceanography Centre Southampton) and lies outside the scope of the present study (see Storkey et al., 2018 for evaluation of its immediate predecessor configuration). Specific to this study, the volume transports provided in Figure 1c have been compared with earlier studies on line no: 47 - 51. Qualitatively we have also validated the transport pathways with existing literature on line no: 55 - 62. The AMOC lower limb transport and the density of the AMOC has been compared with observational evidence on line no: 99 - 102.

Storkey, D., Blaker, A. T., Mathiot, P., Megann, A., Aksenov, Y., Blockley, E. W., Calvert, D., Graham, T., Hewitt, H. T., Hyder, P., Kuhlbrodt, T., Rae, J. G. L., and Sinha, B.: "UK Global Ocean GO6 and GO7: a traceable hierarchy of model resolutions", *Geoscientific Model Development*, 11, 3187-3213, <https://doi.org/10.5194/gmd-11-3187-2018>, 2018.

Please provide the uncertainty of transport listed in Figure 1c.

Answer : Thank you for the suggestion. The uncertainty of transport can arise from various reasons e.g., the ocean model used, the surface forcing dataset, the resolution of the model and the forcing product, initial condition etc. While it

would be instructive to evaluate transports and processes in other simulations, we do not have access to additional high-fidelity hindcasts, and such an inter-comparison exercise is anyhow beyond the scope of the current study. We further recognise that uncertainty may arise from details of the Lagrangian calculation, such as the maximum volume transport assigned to each trajectory. We have now performed a sensitivity analysis and mentioned it on line no: 206 - 208 that “A sensitivity test was performed through varying the maximum volume transport by $\pm 500 \text{ m}^3 \text{ s}^{-1}$ for the two Lagrangian experiments. The results remained unchanged within 1% uncertainty level”.

Why not show the along-pathway water mass properties of Atlantic water run? Perhaps warrant a supplementary figure.

Answer : Thank you for this constructive advice. The along-pathway water mass properties of Atlantic water run is now provided in Supplementary Fig. 1.

Please consider labelling some of streamfunction transport numbers in Figure 3a-c

Answer : We have now modified the Figure 3c and included the Atlantic and Arctic water contribution (transport) to the AMOC lower limb. The Figure 3 caption also mentioned this “...The Atlantic and Arctic water contribution to the AMOC lower limb are shown with bold orange and blue numbers respectively...”.

Why there is a small overturning cell (0.7 Sv) for the dense portion of Atlantic water? Please explain the potential process

Answer : The transformation of very dense portion of Atlantic water is corresponds with the mixing processes and now mentioned on line no: 96 - 98 as “A small overturning cell is also present in density space with an amplitude of 0.7 Sv in which Atlantic waters with densities higher than 27.89 kg m^{-3} overturn, becoming waters with densities between 27.89 kg m^{-3} and 27.57 kg m^{-3} (Figure 3c, clockwise rotating cell) due to mixing”.

Please elaborate the used Eulerian surface water mass transformation framework. In the Eulerian framework, how much of the Atlantic Waters are transformed due to air-sea interactions in the region south/north of 70°N

Answer : The mathematical equations to compute the Eulerian surface water mass transformation are now provided (equations 8-10). The Atlantic Water transformation due to air-sea interactions in the region south of 70°N is mentioned on line no: 109 -112 as "The Eulerian surface watermass transformation framework (calculated using equation 10 and between $\approx 60^\circ\text{N}$ and 70°N) revealed that 7.2 Sv of Atlantic waters (52% of total AMOC and 67% of Atlantic waters overturning) are transformed to lower limb (densities $>27.57 \text{ kg m}^{-3}$), due to air-sea interactions".

Reply to reviewer 2 *Thank you for the response on our article. We are grateful for all your constructive suggestions, which have helped us improving the manuscript. Below you can see our answers. The line numbers are from the revised manuscript*

Major Comments

The headline result is that the AMOC lower limb consists of “72% of Atlantic and 28% of Arctic waters” (Abstract). However, this result depends on the time horizon of the Lagrangian particle integration. In this case the time horizon is 1979-2021 (43 years, from Page 6 on Methods and Data). The question is: How do the 72%/28% results depend on this time horizon?

Answer : We have chosen 1979 - 2021 time horizon because the current study is focused on present-day climate. This is now clarified on line no: 41 and also on line no: 195 - 197 as “In both simulations trajectories were seeded across all 12 months of the year 1979 and advected forward in time for 43 years, i.e., until 2021 to represent the present-day climate”. We have now added on line no: 197 -199 that “Note that the original hindcast model data is available from 1958 to 2021 but we discarded the first 21 years to minimize the effect of model drift on the results. Additionally, the JRA55-do atmospheric forcing dataset uses satellite products for bias-correction and thus more realistic 1979 onward”.

In addressing this question, a couple of conceptual issues occur: The flow is recirculating, so on a long-enough timescale all water comes from everywhere else. Similarly, on a short-enough timescale the AMOC lower limb consists of zero percent Atlantic water and zero percent Arctic water. Specifically, what the authors call “Arctic” water consists of older “Atlantic” water that’s recirculated through the Arctic Ocean. The authors acknowledge this point at the top of page 6. E.g., on a longer time horizon, the orange Atlantic streamlines in Fig. 3 connect to the

dashed blue Arctic streamlines coming from 80N. The present particle experiments eliminate this connection from the outset. Therefore, the 72% / 28% split isn't a very meaningful result. A more robust diagnostic of the connectivity from the eSPNA to the AMOC lower limb is to look at the *distribution* of Lagrangian particle time horizons from the Atlantic and Arctic release lines to the AMOC lower limb (and also from the Atlantic release line to the Arctic release line).

Answer : Thank you for raising this conceptual point. It is important to note that water parcels need time to travel between two sections. The near surface parcels will travel faster than the deeper water parcels and thus the timescales will be different. In the present study we have focused on understanding formation of the total AMOC lower limb, through all light-to-dense water mass transformation north of the eastern SPNA section. We have noted that 43 years is also not long enough for all the Arctic outflow trajectories to reach any of the defined ending sections. This is now clarified on line no: 211 -213 as "Trajectories exiting through the sea surface (i.e. evaporating) and the trajectories still circulating without reaching any of the ending sections were discarded. A total of 0.02 Sv evaporated and 0.62 Sv is still circulating in the Arctic outflow simulation (Figure 1c)". If we would have taken a shorter time horizon, then more trajectories would have circulated without reaching ending sections and our estimate would have been more uncertain. In that case, we would have been more unsure whether the circulating parcels transform to AMOC lower limb and exit the eastern SPNA southward or not. We had to stop at year 2021 as there are no more data available further for this particular hindcast. The "Atlantic" and "Arctic" water means water parcels that are arriving from the Atlantic and Arctic Ocean and now mentioned on line no: 38 - 40 as "It is important to note that the "Atlantic" and "Arctic" water denote water parcels that are arriving from the Atlantic and Arctic Ocean after upstream property modification". This is also the traditional way to define a water mass, by local contrast with surrounding waters and preceding transformation. During the transit through the Atlantic and Arctic, the water parcels change their

properties and thus in Figure 3a-c one can notice that the starting temperature, salinity and density are different for Atlantic and Arctic waters.

The computation of overturning stream functions from the Lagrangian particle results using TRACMASS (Fig 3) is interesting, but I question its validity. Specifically: How does a Lagrangian particle retain its volume transport tag (F_m in equation (2))? This idea is essential to construct the overturning stream functions. For example, on page 7 it says “to each trajectory simulated by TRACMASS a volume transport was assigned, which is conserved throughout its journey due to the volume conserving algorithm.” There are several unstated assumptions here, which need to be stated explicitly and justified. I’m skeptical because I expect nearby particles to diverge exponentially (at least asymptotically, for a while) due to Lagrangian chaotic advection. This is called stirring in the sense of Eckart (1948, J. Mar. Res.); see also LaCasce & Bower (2000, J. Mar. Res.). That means there will be divergence of the ensemble of particles initially tagging the water associated with a particular F_m . Therefore, I don’t understand how a single particle can be associated with a particular F_m . How is precipitation and evaporation handled? Do particles enter and leave the surface ocean? Or adjust their F_m ? This issue undermines the stream functions in Fig. 3 and the heat/salt divergences in Fig. 4. In short, I’m worried that the Fig. 3, 4 results aren’t robust.

Answer : We realised that we should have explained better about the working principle of TRACMASS which relies on mass/volume conservation. We have now added few sentences about how to interpret a TRACMASS simulated trajectory on line no: 184 - 189 as “A trajectory calculated by TRACMASS can be viewed as a streamtube. In the absence of stochastic parameterisation, the volume transport at one end of the streamtube must be equal to the volume transport at the other end (non-divergent). Thus, the streamtubes are equivalent to the volume transport pathways of an incompressible fluid. Sub-grid scale processes are not

parameterized during the TRACMASS analyses. Parcel properties (temperature, salinity) do however evolve along trajectories, as a consequence of parameterized mixing and air-sea interaction in the hindcast model". A full description of how TRACMASS conserves mass/volume along trajectories is provided in Döös et al., 2017. The way to interpret trajectories calculated from different Lagrangian models is discussed in Van Sebille et al., 2018. The approach to compute streamfunctions based on mass conserving trajectories is mature after just over 20 years, since early such streamfunctions were first published (e.g., Speich et al. 2001). In these early studies, trajectories were calculated with steady state or repeat-year currents and properties until all particles exited a region. The present trajectories are subject to time-dependent model fields over the hindcast era and trajectories are hence limited to a maximum time span of up to 43 years. The density-latitude streamfunction obtained from Lagrangian trajectories in Figure 3c can also be visually compared with earlier published streamfunctions (e.g., Roberts et al., 2020) calculated from Eulerian viewpoint. In the present study, the trajectories exiting through the sea-surface have been discarded and now mentioned on line no: 211 - 213 as "Trajectories exiting through the sea surface (i.e. evaporating) and the trajectories still circulating without reaching any of the ending sections were discarded. A total of 0.02 Sv evaporated and 0.62 Sv is still circulating in the Arctic outflow simulation (Figure 1c)".

Döös, K., Jönsson, B. and Kjellsson, J. Evaluation of oceanic and atmospheric trajectory schemes in the tracmass trajectory model v6.0. *Geosci. Model. Dev.* 10, 1733-1749 (2017)

Van Sebille, E. et al. Lagrangian ocean analysis: Fundamentals and practices. *Ocean. Model.* 121, 49-75 (2018)

Speich, S., Blanke, B., and Madec, G. Warm and cold water routes of an OGCM thermohaline conveyor belt. *Geophys. Res. Lett.* 28, 311-314 (2001)

Roberts, M. J. et al. Sensitivity of the atlantic meridional overturning circulation to model resolution in cmip6 highresmip simulations and implications for future

changes. *J. Adv. Model. Earth Syst.* 12, e2019MS002014 (2020)

Minor Comments

Abstract (first sentence): It says “Deep-water formation in the eastern Subpolar North Atlantic Ocean (eSPNA) and Nordic Seas is crucial for maintaining the lower limb of the Atlantic Meridional Overturning Circulation (AMOC) and stabilizing Earth’s climate.” In what sense is the AMOC “stabilizing” of Earth’s climate? Also, deep water formation occurs in the western SPNA. Why isn’t that mentioned? I suggest this sentence is rewritten and/or better qualified.

Answer : The first sentence of the abstract has now been modified as “Deep-water formation in the eastern Subpolar North Atlantic Ocean (eSPNA) and Nordic Seas is crucial for maintaining the lower limb of the Atlantic Meridional Overturning Circulation (AMOC), of consequence for global climate”. The consequential role of AMOC in the Earth’s climate is mentioned on line no: 14 - 21. Additionally, observational evidence suggests that deep water formation north of OSNAP-East is important for the AMOC lower limb transport. This is also stated on line no: 22 - 24 as “A trans-basin observational array (OSNAP) showed that the lower limb preferably forms north of the OSNAP-East section, which extends from the south-eastern tip of Greenland to Scotland”. However, we cannot mention all of these details in the abstract, within the 150-word limit. The deep water formation in the western SPNA is important for downward transport of heat, nutrients etc. but not for majority of the time-mean AMOC lower limb transport.

Abstract (final sentence): It says “The Atlantic waters that gain density by air-sea fluxes along the eastern periphery of Nordic Seas and do not enter the Arctic Ocean are not critical in determining the time-mean AMOC strength.” This is confusing. Why should these Atlantic waters determine the AMOC “strength” (i.e., transport)? The authors

expect something that's not clear to me here. Also, the penultimate line on page 1, and the last line on page 5 refer to the "AMOC strength" this way.

Answer : A majority of the earlier studies have reported that along the eastern periphery of Nordic Seas, the low density Atlantic water get transformed to high density deep water due to air-sea fluxes. We did mention that on line no: 26 - 28 and line no: 138 - 139. The rate of deep water formation is generally equated to an AMOC strength. However, those studies were unable to explain through which pathways the newly formed dense Atlantic water leaves the eastern SPNA. We have shown in the present study that if the air-sea flux is important for AMOC lower branch, the high density deep Atlantic waters along the eastern rim of Nordic Seas have to enter the Arctic Ocean and then exit the eastern SPNA. This is now mentioned on line no: 143 -148 as "However, ocean heat loss along the Nordic Seas periphery can have an indirect effect on the Arctic outflow. After losing heat to the atmosphere, a substantial fraction of densified Atlantic waters travel further northward and enter the Arctic Ocean through Fram Strait and the Barents Sea. In the Arctic, these Atlantic waters may be further modified through interaction with the atmosphere and Arctic freshwater before being exported southward again through Fram Strait. An extension of the present study could thus be to trace Atlantic waters through the Arctic Ocean. This will help to identify and quantify the water mass transformation of Atlantic waters in the Arctic, of indirect consequence for the AMOC".

Introduction (page 1): It says "Dense waters from the Nordic Seas are transported across the Greenland-Scotland Ridge (GSR) to the eSPNA by the East Greenland Current (EGC)". This is true, but dense waters also cross the GSR east of Iceland, for example, in the Faroe Bank Channel.

Answer : Thank you for raising this point. We have now mentioned about the Iceland-Faroe Slope Jet on line no: 31 which takes dense water towards the Faroe

Bank Channel.

Introduction (page 1): Clarify where it says “The other half was found to be associated with oceanic heat loss in the Irminger Sea and Iceland basin.”

Answer : We had cited Petit et al. (2020) on line no: 32 and line no: 34. In that study, the authors quantified the deep water formation. They reported that 6.6 ± 0.4 Sv of dense water is transported across the GSR and the other half (i.e. 6.6 ± 3.8 Sv out of 13.2 ± 3.8 Sv) is found to be formed at the Irminger Sea and Iceland basin due to the air-sea fluxes.

Fig. 1: Explain what the colors on the particle trajectories mean. The Barents Sea section is at an unusual angle. Usually, it joins Svalbard to Norway via Bear Island (e.g., Tsubouchi et al., 2021, 10.1038/s41558-020-00941-3).

Answer : We have now clarified in Figure 1 caption that “The pathways of each trajectory are marked with different colors.” We agree that the Barents Sea section is at an unusual angle but it will not affect the results presented in the study. This is since we just wanted to stop the trajectories to go into the Arctic Ocean due to the longer travel timescale and data unavailability after 2021.

Fig. 2: What depth is used for the maps in c, d, e, and h?

Answer : Thank you for pointing towards this. We have now stated on Figure 2 caption that “All depth levels, associated with trajectories, are considered in this computation”.

Fig. 3/page 4: It says that Fig. 3d verifies the 10.8 Sv overturning value. How could Fig. 3d not “verify” the 10.8Sv? Fig. 3d is just a slice of stream function at the left edge of Fig. 3c. It seems to me that it must, by construction, give a consistent total transformation rate. In other words, I don’t think Fig. 4d is an independent verification of the 10.8Sv num-

ber.

Answer : We completely agree with your point. Figure 3d and the associated discussion is now removed from the revised manuscript.

Fig. 4/page 5: In the text it says that 67% of Atlantic waters are transformed by air-sea interaction and 33% are transformed by interior mixing. But Fig. 4 (bottom row) shows that most Atlantic water transformation occurs beneath the mixed layer (the blue line is very similar to the black line). These results contradict each other. Please explain and reconcile the contradiction.

Answer : Thank you for raising this important point. We have now clarified with an example (Example Figure 1) and supplementary figures on line no: 119 - 127 that below the MLD heat divergence does not necessarily result from interior mixing. Line no: 119 - 127 states that “It is worth noting that Atlantic waters are also losing heat below the MLD downstream of Denmark Strait (Supplementary Fig.3). However, the Arctic waters are not gaining heat at these locations (Supplementary Fig.2). This indicates that the heat loss of Atlantic waters at these locations is occurring just below the base of the MLD and is due to the air-sea fluxes. For an example, let us consider a winter-time trajectory within the MLD enters a grid box through its western grid wall, moves just below the MLD and exits through the eastern wall. The trajectory will thus experience the winter-time cooling before it leaves the grid box. When calculating the heat divergence at the exit wall (or in other words below the MLD), a cooling of the parcel will be observed. Thus, although the total heat divergence obtained from the Atlantic inflows is mostly determined by the below MLD heat divergence (Figure 4, bottom panel), it should be considered as a result of mixing if the Arctic waters are gaining heat at the same locations”.

Example Figure 1: A sketch to illustrate how a water parcel loses heat just at the base of the Mixed layer depth (MLD). Green Arrow indicates a trajectory pathway.

Page 6: When talking about future AMOC slowdown it says: “Reduced sub-surface mixing of Atlantic waters is a likely part of overall AMOC slowdown”. What’s the evidence to support this claim?

Answer : We have modified few sentences on line no: 152 - 158 as “Global climate model simulations robustly indicate a slowdown of the AMOC in warmer future climate due to the freshwater induced enhancement in surface stratification. We have emphasised here the importance of mixing between Atlantic and Arctic waters, as a substantial component of water mass transformation, necessary to sustain the lower limb and to achieve the density and depth of the return flow. By inhibiting mixing through raised potential energy demand, stronger stratification will reduce the amount of turbulent kinetic energy available for this mixing. Reduced sub-surface mixing of Atlantic waters is thus potentially the key to overall AMOC slowdown,..”.

Page 6, final sentence of main text: It says “The proposed importance of interior mixing can also be helpful to explain why some of the climate models that underestimates the AMOC strength also have shallower AMOC.” This needs to be clarified and explained better.

Answer : We have now modified the sentence on line no: 149 - 152 as “Given the importance of interior mixing in sustaining a realistic-strength AMOC in the current hindcast, we note that some HighResMIP (High Resolution Model Intercomparison Project) climate models underestimating AMOC strength are characterised by a shallow lower limb. This highlights the importance of correctly

representing both water mass properties (densities), pathways, and transformation processes (surface and interior) in climate models”.

REVIEWER COMMENTS

Reviewer #1 (Remarks to the Author):

I thank the authors for their response to my questions and comments. On reading the revised manuscript, I do however have one further major comment:

The meridional overturning streamfunction in Figure 3c suggests that the Arctic water contributes to 4.3 Sv of the lower limb of AMOC. It also shows existence of most of Arctic water (3-4 Sv with density > 27.5) across Fram Strait at 80 °N. I wonder where and how this dense Arctic water was formed? Previous study (e.g., Mauritzen 1996) suggested that this dense water can be related to the transformation firstly along the eastern periphery of Nordic Seas and then along the boundary of Arctic. A recent study (Årthun 2023) shows dominant role of the former using the Eulerian framework. However, the authors argued that they did not consider the Atlantic water trajectories that enter the Arctic Ocean (about 6.6 Sv from Figure 1c). It's essential to prove and show how much of the missing 6.6 Sv Atlantic water can be transformed within the Nordic Seas. The authors need to address this before they concluded a negligible contribution of densification along the eastern periphery of Nordic Sea to the lower limb of AMOC.

Other minor comments:

1) In addition to the heat and salt divergence as a function of Latitude, I would suggest a table that lists the Lagrangian heat and salt divergence together with Eulerian air-sea heat and freshwater fluxes for the regions north/south of Greenland Scotland Ridge. This can clearly show the relative contribution of different processes, water masses and regions.

2) I would suggest a supplementary figure showing the lateral distribution of air-sea heat/freshwater fluxes that the Atlantic/Arctic trajectories experienced, as well their ratio to the depth integrated Lagrangian heat and salt divergence.

References:

Mauritzen, C. (1996). Production of dense overflow waters feeding the North Atlantic across the Greenland-Scotland Ridge. Part 1: evidence for a revised circulation scheme. *Deep Sea Res.* 43, 769–806.

Årthun, M. (2023). Surface-forced variability in the Nordic Seas overturning circulation and overflows. *Geophysical Research Letters*, 50, e2023GL104158.

Reviewer #2 (Remarks to the Author):

Review of revision of: "Formation of the Atlantic Meridional Overturning Circulation lower limb is critically dependent on Atlantic-Arctic mixing" by Dey et al. for Nature Communications

Overall:

The authors have adequately addressed most of the minor comments from the first review (with a couple of exceptions that still need fixing, see below). They have clarified major comment 2 about attributing transport values to Lagrangian parcels ("particle" has been replaced everywhere with "parcel" and a discussion on stream tubes is added). And they've rebutted major comment 1 about sensitivity of the headline 72%/28% result on the 43 year parcel time horizon.

The paper is improved because it's clearer. My major concerns from the first version have been superficially addressed, however (especially the first one). These concerns are about the main results of the paper (e.g., the title and abstract). It's hard to know how the main results will be affected by thoroughly fixing the concerns (especially the first one). Therefore, I still recommend the paper is declined and I encourage the authors to resubmit a new manuscript that's thoroughly rethought.

Comments on rebuttal to major comments:

1: In defending the choice of 43 years, the authors say "If we would have taken a shorter time horizon, then more trajectories would have circulated without reaching ending sections and our estimate would have been more uncertain. In that case, we would have been more unsure whether the circulating parcels transform to AMOC lower limb and exit the eastern SPNA southward or not. We had to stop at year 2021 as there are no more data available further for this particular hindcast." In other words, the choice of 43 years is a balance between accuracy (not too short) and practicality (can't go past 2021).

There's still a conceptual mistake here. The mistake is that a single time horizon (the 43 years) can capture the Arctic-Atlantic mixing and the AMOC lower-limb sources. Instead, the mixing and sources are characterized by a *distribution* over time horizons. I.e., the 72%/28% split depends on time horizon. That's why the 72%/28% isn't very significant: it would be different if it was some other time than 43 years. I made these points in the first review, but perhaps it was unclear (for more background see Holzer & Primeau 2006, GRL; Shah et al. 2017, JPO; Holzer et al. 2020, JGR). The robust way to quantify the upstream sources of the AMOC lower limb is to compute backwards Lagrangian trajectories starting from the AMOC lower limb in 2021. Then start the analysis by plotting the *distribution* of arrival times at the upstream (Arctic, Atlantic) sections.

I'm not saying the present results are irrelevant, just that they're not robust. By robust I mean that it's hard to reach a robust interpretation/conclusion with these results on their own, not that the results are wrong. The final three sentences of the abstract obscure this issue. E.g., "43 years"

doesn't appear, and saying that the eastern Nordic Seas have a "negligible contribution" to the AMOC lower limb suggests that the time horizon is irrelevant, which isn't true. This is misleading and/or misinterprets the main results.

2: The authors say "A trajectory calculated by TRACMASS can be viewed as a streamtube...the volume transport at one end of the streamtube must be equal to the volume transport at the other end." They provide some more discussion and clarification, which is useful. The van Sebille et al. (2017) reference (their section 2.4 on streamtubes) is most relevant (also lines 184–187, which are new). An assumption for the stream tube idea to work is that the flow field is steady (van Sebille et al, 2017, section 2.4). That assumption is not true in this case, however, although the authors don't mention it (they should).

In any case, even for steady flow, I'm skeptical that with chaotic advection the idea will work: Lagrangian trajectories typically diverge exponentially, regardless of how close to each other they start. Eventually, their trajectories become uncorrelated. So an initial tube of particles eventually becomes arbitrarily distorted. How can one trajectory from that tube characterize the original tube transport?

In this argument about stream tubes eventually falling apart, the time taken for trajectories to become uncorrelated is critical (i.e., the time for Lagrangian chaos to set in). Is that timescale greater than 43 years here? If so, the stream tube idea should work (because Lagrangian chaos hasn't begun). Looking at the trajectories in Fig. 1 it seems unlikely, although it's hard to tell because these are just a few randomly picked trajectories.

Here's a test to resolve the question. It's motivated by the final paragraph of section 3.2.3 in van Sebille et al., 2018, starting "Streamtube-based volume transport is reversible, so that backward integrations can be performed to track the origin of a given volume". Compute backward Lagrangian trajectories starting from the AMOC lower limb in 2021 (as also suggested above). Then, for the trajectories that cross the upstream (Arctic, Atlantic) sections, check that the trajectories are reversible and the stream tubes connecting the start and end locations are intact. I.e., calculate the transport arriving at the Arctic lower limb from the upstream sections using both the forward and backward trajectories. They should agree if the stream tube idea works.

Comments on rebuttal to Minor comments:

The Fig 4/page 5 issue on 67% of Atlantic waters are transformed by air-sea interaction: The authors say that even though the transformation is below the mixed layer base, it's still due to air-sea interaction (see Example Figure 1 in the rebuttal). I agree, that's a plausible mechanism. But it's not the only plausible mechanism. E.g., Atlantic waters below the mixed layer can be transformed by internal diabatic processes unrelated to air-sea interaction (or, mixing with Arctic water, for that matter, see line 124). To tell if the transformation is due to air-sea interaction, compute the transformation that occurs while the parcel is in the mixed layer. Incidentally, the new supplementary figures on transformation are useful.

The page 6 future AMOC slowdown issue: The new text says “Global climate model simulations robustly indicate a slowdown of the AMOC in warmer future climate due to the freshwater induced enhancement in surface stratification (Cheng et al. 2013, Liu et al. 2017, Weijer et al. 2020).” The final phrase starting “due to ...” is new. The explanation for AMOC weakening (freshwater-induced stratification enhancement) is misleading, however. From my reading, none of the papers cited make that claim and it’s still an open question why climate models project an AMOC slowdown. This sentence needs to be rephrased to be more accurate.

Reply to reviewer 1 *Thank you for the response on our article. We are grateful for all your constructive suggestions, which have helped us improving the manuscript. Below you can see our answers. The line numbers are from the revised manuscript*

Major Comments

The meridional overturning streamfunction in Figure 3c suggests that the Arctic water contributes to 4.3 Sv of the lower limb of AMOC. It also shows existence of most of Arctic water (3-4 Sv with density >27.5) across Fram Strait at 80°N. I wonder where and how this dense Arctic water was formed? Previous study (e.g., Mauritzen 1996) suggested that this dense water can be related to the transformation firstly along the eastern periphery of Nordic Seas and then along the boundary of Arctic. A recent study (Árthun 2023) shows dominant role of the former using the Eulerian framework. However, the authors argued that they did not consider the Atlantic water trajectories that enter the Arctic Ocean (about 6.6 Sv from Figure 1c). It's essential to prove and show how much of the missing 6.6 Sv Atlantic water can be transformed within the Nordic Seas. The authors need to address this before they concluded a negligible contribution of densification along the eastern periphery of Nordic Sea to the lower limb of AMOC.

Answer : We agree with your viewpoint and realised we should have explained better in our previous submitted manuscript regarding the contribution of densification along the eastern periphery of the Nordic Seas to the lower limb of AMOC. You are right that 6.6 Sv of Atlantic waters reaches the Arctic Ocean and can contribute to the lower branch of AMOC through Fram strait outflow. We also think this is happening but it seems we were unable to get the message through our writing. We have now modified the last sentence of the abstract as “In contrast, Atlantic waters gaining density through air-sea interaction along the eastern periphery of Nordic Seas and not entering Arctic Ocean make a negligible contribu-

tion to the lower limb". We had discussed elaborately your points on line no: 142 - 143 "We have also shown that low-to-high density water mass transformation of Atlantic waters (that are not entering the Arctic Ocean) along the eastern rim of the Nordic Seas is only a minor component of the AMOC" , line no: 146 - 149 "We show that neither open ocean convection in the Greenland and Iceland Seas, mixing between gyre and boundary, nor heat loss of ocean waters along the eastern periphery of Nordic Seas (or in other words along the NwAFC and Norwegian Atlantic Slope Current) which do not enter the Arctic Ocean are sufficient to transform waters to the high densities necessary to sustain the AMOC lower limb" and on line no: 149 - 155 "However, ocean heat loss along the Nordic Seas periphery can have an indirect effect on the Arctic outflow. After losing heat to the atmosphere, a substantial fraction of densified Atlantic waters travel further northward and enter the Arctic Ocean through Fram Strait and the Barents Sea. In the Arctic, these Atlantic waters may be further modified through interaction with the atmosphere and Arctic freshwater before being exported southward again through Fram Strait. Such a scenario would be in line with previous findings. An extension of the present study could thus be to trace Atlantic waters through the Arctic Ocean, on a longer timescale. This will help to identify and quantify the water mass transformation of Atlantic waters in the Arctic, of indirect consequence for the AMOC". We hope these clarifications will make things clearer. In a nutshell, the densification along the eastern periphery of the Nordic Seas is important but the waters that are gaining density there must transit through the Arctic Ocean in order to contribute to the AMOC lower limb.

Other Minor points

In addition to the heat and salt divergence as a function of Latitude, I would suggest a table that lists the Lagrangian heat and salt divergence together with Eulerian air-sea heat and freshwater fluxes for the regions north/south of Greenland Scotland Ridge. This can clearly show the relative contribution of different processes, water masses and regions.

Answer : We appreciate your constructive suggestion. Note that, the Lagrangian heat and salt divergences presented here are obtained only from those water trajectories that have reached the eSPNA section as the focus of the present study is on AMOC lower limb. A direct comparison between Eulerian air-sea fluxes and Lagrangian heat and salt divergences would have been possible if we have taken into account all the water-masses in the region and constructed heat and salt budgets as done by Berglund et al., 2023. The computation of heat and salt budget over the Nordic Seas lies outside the study objectives.

Berglund, S., Döös, K., Groeskamp, S. and McDougall, T. North Atlantic Ocean circulation and related exchange of heat and salt between water masses. *Geophysical Research Letters*. 50, e2022GL100989 (2023)

I would suggest a supplementary figure showing the lateral distribution of air-sea heat/freshwater fluxes that the Atlantic/Arctic trajectories experienced, as well their ratio to the depth integrated Lagrangian heat and salt divergence.

Answer : Thank you for the suggestion. The tracer equation for conservation of temperature (T) and salinity (S) can be described as

$$\frac{DT}{Dt} = D_T + Q, \quad (1)$$

$$\frac{DS}{Dt} = D_S + (E - P - R). \quad (2)$$

Here, D_T is the diffusion of temperature, Q represents net heat flux, D_S is diffusion of salinity and $(E - P - R)$ is freshwater flux. Expanding the derivatives in Equation 1 and 2 along with incompressibility assumption and presuming the long term mean of local rate of change is small compared to the advective rate, the

equations can be reduced to

$$\nabla \cdot (VT) = D_T + Q, \quad (3)$$

$$\nabla \cdot (VS) = D_S + (E - P - R), \quad (4)$$

where V is the three dimensional velocity field. These are all described elaborately in Berglund et al., 2017, which is the first study computing Lagrangian heat and salt divergences using TRACMASS. Here, $\nabla \cdot (VT)$ and $\nabla \cdot (VS)$ are the heat and salt divergences respectively. In the present study, these divergence quantities have been computed using only those water trajectories that reached the eSPNA section. The Eulerian net heat and freshwater flux affect all the near-surface waters in a region. However, it is not possible to determine the distribution of the total Eulerian fluxes among different water-masses. If we would have taken all the water-masses in the region then we could have used equations 3 and 4 to construct a map that shows the ratio between depth integrated divergence and air-sea flux. We have reported on line no: 109 -112 that “The Eulerian surface watermass transformation framework (calculated using equation 10 and between $\approx 60^\circ\text{N}$ and 70°N) revealed that 7.2 Sv of Atlantic waters (52% of total AMOC and 67% of Atlantic waters overturning) are transformed to lower limb (densities $>27.57 \text{ kg m}^{-3}$), due to air-sea interactions”. This is solely due to the fact that the meridional overturning streamfunction in density-latitude frameowrk shows the Arctic waters transform due to mixing only, while the Atlantic waters change properties due to air-sea fluxes and mixing. This is discussed elaborately in the results section and also on line no: 105 - 109 as “The meridional overturning streamfunction in density-latitude coordinates (Figure 3c) shows that the AMOC lower limb formation is mostly located between $\approx 60^\circ\text{N}$ and 70°N , and that Arctic waters contribute to the AMOC lower limb through changes in temperature, salinity and density from Denmark Strait onwards, where they encounter Atlantic in-flows. Interior mixing is found to be the dominant process in transforming Arctic waters that join the AMOC lower limb, while Atlantic waters are transformed

through a combination of air-sea fluxes and interior mixing”.

Berglund, S., Döös, K. and Nycander, J. Lagrangian tracing of the water–mass transformations in the Atlantic Ocean. Tellus A: Dynamic Meteorology and Oceanography. 69, 1306311 (2017)

Reply to reviewer 2 *Thank you for the response on our article. We are grateful for all your constructive suggestions, which have helped us improving the manuscript. Below you can see our answers. The line numbers are from the revised manuscript*

Major Comments

In defending the choice of 43 years, the authors say "If we would have taken a shorter time horizon, then more trajectories would have circulated without reaching ending sections and our estimate would have been more uncertain. In that case, we would have been more unsure whether the circulating parcels transform to AMOC lower limb and exit the eastern SPNA southward or not. We had to stop at year 2021 as there are no more data available further for this particular hindcast." In other words, the choice of 43 years is a balance between accuracy (not too short) and practicality (can't go past 2021).

There's still a conceptual mistake here. The mistake is that a single time horizon (the 43 years) can capture the Arctic-Atlantic mixing and the AMOC lower-limb sources. Instead, the mixing and sources are characterized by a *distribution* over time horizons. I.e., the 72%/28% split depends on time horizon. That's why the 72%/28% isn't very significant: it would be different if it was some other time than 43 years. I made these points in the first review, but perhaps it was unclear (for more background see Holzer & Primeau 2006, GRL; Shah et al. 2017, JPO; Holzer et al. 2020, JGR). The robust way to quantify the upstream sources of the AMOC lower limb is to compute backwards Lagrangian trajectories starting from the AMOC lower limb in 2021. Then start the analysis by plotting the *distribution* of arrival times at the upstream (Arctic, Atlantic) sections.

I'm not saying the present results are irrelevant, just that they're not robust. By robust I mean that it's hard to reach a robust interpreta-

tion/conclusion with these results on their own, not that the results are wrong. The final three sentences of the abstract obscure this issue. E.g., “43 years” doesn’t appear, and saying that the eastern Nordic Seas have a “negligible contribution” to the AMOC lower limb suggests that the time horizon is irrelevant, which isn’t true. This is misleading and/or misinterprets the main results.

Answer : We appreciate your concerns and realised we did not fully understand your question in the previous review. We agree that the water trajectories will experience different air-sea flux and mixing conditions depending on the time horizon. However, we have expected that the time-mean AMOC strength will not change significantly as the study focuses on averaging air-sea fluxes and mixing experienced by the trajectories over a timescale of 43 years. To test the robustness of the results, we have now back-traced southward moving waters from the eSPNA section, as suggested. The results remain very similar and mentioned on line no: 128 - 133 as “We found that the results hold when back-tracking the southward flowing ocean waters from the eSPNA section for the year 2021. The meridional overturning stream function in density-latitude coordinate and the contributions of the Atlantic and Arctic waters to the AMOC lower limb from the backward tracking experiment (Supplementary Fig.4) shows the results are very similar to those for the forward run. This strengthens our conclusions and shows robustness of the results. However, backward tracing is insufficient to quantify the total amount of transport from the Fram Strait and eSPNA sections, and respective partitioning, compared to the forward trajectories”. We somewhat disagree on the point that “ The robust way to quantify the upstream sources of the AMOC lower limb is to compute backwards Lagrangian trajectories starting from the AMOC lower limb in 2021”. This is since the waters that are flowing southward across the eSPNA section have to come from the Fram Strait and eSPNA section and thus if we do a forward and backward experiment the results would be similar. We realise that we were not very clear in our previous submitted manuscript regarding the contribution of densification along the eastern periph-

ery of Nordic Sea to the lower limb of AMOC. We have now modified the last sentence of the abstract as “In contrast, Atlantic waters gaining density through air-sea interaction along the eastern periphery of Nordic Seas and not entering Arctic Ocean make a negligible contribution to the lower limb”. The role of water densification along the eastern part of the Nordic Seas on the AMOC has been discussed elaborately on line no: 142 - 143 “We have also shown that low-to-high density water mass transformation of Atlantic waters (that are not entering the Arctic Ocean) along the eastern rim of the Nordic Seas is only a minor component of the AMOC” , line no: 146 - 149 “We show that neither open ocean convection in the Greenland and Iceland Seas, mixing between gyre and boundary, nor heat loss of ocean waters along the eastern periphery of Nordic Seas (or in other words along the NwAFC and Norwegian Atlantic Slope Current) which does not enter the Arctic Ocean are sufficient to transform waters to the high densities necessary to sustain the AMOC lower limb” and on line no: 149 - 155 “However, ocean heat loss along the Nordic Seas periphery can have an indirect effect on the Arctic outflow. After losing heat to the atmosphere, a substantial fraction of densified Atlantic waters travel further northward and enter the Arctic Ocean through Fram Strait and the Barents Sea. In the Arctic, these Atlantic waters may be further modified through interaction with the atmosphere and Arctic freshwater before being exported southward again through Fram Strait. Such a scenario would be in line with previous findings. An extension of the present study could thus be to trace Atlantic waters through the Arctic Ocean, on a longer timescale. This will help to identify and quantify the water mass transformation of Atlantic waters in the Arctic, of indirect consequence for the AMOC”. We hope these clarifications will make things clearer. In a nutshell, the densification along the eastern periphery of the Nordic Seas is important but the waters that are gaining density there must transit through the Arctic Ocean in order to contribute to the AMOC lower limb.

The authors say “A trajectory calculated by TRACMASS can be viewed as a streamtube...the volume transport at one end of the streamtube must

be equal to the volume transport at the other end." They provide some more discussion and clarification, which is useful. The van Sebille et al. (2017) reference (their section 2.4 on streamtubes) is most relevant (also lines 184-187, which are new). An assumption for the stream tube idea to work is that the flow field is steady (van Sebille et al, 2017, section 2.4). That assumption is not true in this case, however, although the authors don't mention it (they should).

In any case, even for steady flow, I'm skeptical that with chaotic advection the idea will work: Lagrangian trajectories typically diverge exponentially, regardless of how close to each other they start. Eventually, their trajectories become uncorrelated. So an initial tube of particles eventually becomes arbitrarily distorted. How can one trajectory from that tube characterize the original tube transport?

In this argument about stream tubes eventually falling apart, the time taken for trajectories to become uncorrelated is critical (i.e., the time for Lagrangian chaos to set in). Is that timescale greater than 43 years here? If so, the stream tube idea should work (because Lagrangian chaos hasn't begun). Looking at the trajectories in Fig. 1 it seems unlikely, although it's hard to tell because these are just a few randomly picked trajectories.

Here's a test to resolve the question. It's motivated by the final paragraph of section 3.2.3 in van Sebille et al., 2018, starting "Streamtube-based volume transport is reversible, so that backward integrations can be performed to track the origin of a given volume". Compute backward Lagrangian trajectories starting from the AMOC lower limb in 2021 (as also suggested above). Then, for the trajectories that cross the upstream (Arctic, Atlantic) sections, check that the trajectories are reversible and the stream tubes connecting the start and end locations are intact. I.e., calculate the transport arriving at the Arctic lower limb from the upstream sections using both the forward and backward trajectories. They

should agree if the stream tube idea works.

Answer : Thank you for your thought-provoking comments. In section 3.2.3 of van Sebille et al.(2018), it is clearly explained that the method works with both steady-state and time-evolving fields. In this study, we have used time-evolving fields and sub-sampled them into piecewise steady fields. The piecewise steady solution is also described in van Sebille et al.(2018). It is important to make the distinction between Lagrangian dispersion and fluid divergence. The flow here is non-divergent due to mass conservation and incompressibility. The individual parcels will, however, disperse as they are mixed with other waters. This can be visualised by two water masses merging into each other. The two water masses will typically disperse but each single stream tube (or in other words, parcel) will conserve its mass (equivalently volume) transport. Yes the flow is chaotic but we are making averages over large quantities of trajectories, which sums up as mean flows. Otherwise there would be no mean currents in the ocean transporting heat and salt in and between the oceans. This has been shown in many articles using TRACMASS and Ariane since the 90's (Döös 1995, Blanke et.al., 1997). To test the robustness of the results, we have now back-traced southward moving waters from the eSPNA section, as suggested. The results remain approximately the same and mentioned on line no: 128 - 133 as "We found that the results hold when back-tracking the southward flowing ocean waters from the eSPNA section for the year 2021. The meridional overturning stream function in density-latitude coordinate and the contributions of the Atlantic and Arctic waters to the AMOC lower limb from the backward tracking experiment (Supplementary Fig.4) shows the results are very similar to those for the forward run. This strengthens our conclusions and shows robustness of the results. However, backward tracing is insufficient to quantify the total amount of transport from the Fram Strait and eSPNA sections, and respective partitioning, compared to the forward trajectories.". Note, it is almost impossible (code-wise) to start a trajectory from its ending position after a forward tracking simulation using time-evolving fields. This is since the code (in its current form) cannot start a trajectory from an interpolated time step

– e.g., if a trajectory starts at the Fram Strait in January 1979 (we can do this as we have monthly data from the hindcast model) and reaches the eSPNA section after 11.3 years. Although we know the ending position of that trajectory, we cannot start a backward run from the same location, as the hindcast model (or any model/reanalysis) does not provide data at an interpolated time step (in the above example, we would need data on day 18.25 of March 1990).

Döös, K. Inter-ocean exchange of water masses. *Journal of Geophysical Research: Oceans*. 100(C7), 13499-13514 (1995)

Blanke, B and Raynaud S. Kinematics of the Pacific Equatorial Undercurrent: a Eulerian and Lagrangian approach from GCM results. *Journal of Physical Oceanography*. 27, 1038- 1053 (1997)

Comments on rebuttal to Minor comments:

The Fig 4/page 5 issue on 67% of Atlantic waters are transformed by air-sea interaction: The authors say that even though the transformation is below the mixed layer base, it's still due to air-sea interaction (see Example Figure 1 in the rebuttal). I agree, that's a plausible mechanism. But it's not the only plausible mechanism. E.g., Atlantic waters below the mixed layer can be transformed by internal diabatic processes unrelated to air-sea interaction (or, mixing with Arctic water, for that matter, see line 124). To tell if the transformation is due to air-sea interaction, compute the transformation that occurs while the parcel is in the mixed layer. Incidentally, the new supplementary figures on transformation are useful.

Answer : Separate from air-sea interaction, internal diabatic processes are primarily associated with diapycnal mixing. In the hindcast, this may arise through both entrainment of pycnocline waters into the mixed layer during mixed layer deepening and parameterized mixing processes. The latter encompass vertical and horizontal diffusive mixing and may further lead to density gain (cabbeling)

through the combined mixing of heat and salt. We agree that this may involve mixing with thermocline water masses, distinct from the Arctic outflow. Nevertheless, the evidence from Lagrangian and Eulerian analyses supports our emphasis on the dominant combination of air-sea interaction and mixing with Arctic outflows, in the transformation of Atlantic waters.

The page 6 future AMOC slowdown issue: The new text says “Global climate model simulations robustly indicate a slowdown of the AMOC in warmer future climate due to the freshwater induced enhancement in surface stratification (Cheng et al. 2013, Liu et al. 2017, Weijer et al. 2020).” The final phrase starting “due to ...” is new. The explanation for AMOC weakening (freshwater-induced stratification enhancement) is misleading, however. From my reading, none of the papers cited make that claim and it’s still an open question why climate models project an AMOC slowdown. This sentence needs to be rephrased to be more accurate.

Answer : Thank you for raising this important point and we agree with your statement. The literature (e.g., Cheng et al. 2013, Liu et al. 2017, Weijer et al. 2020) robustly indicate a slowdown of the AMOC in warmer future climate. However, as you have mentioned, the causes of the AMOC slowdown are still debatable. The hypotheses encompass a higher amount of precipitation, increased Greenland ice-sheet melt, enhanced fresh Arctic outflow, and availability of more warm water at the deep convection site. These all lead to reduced surface density and enhanced stratification. We now address this on line no : 159 - 166, specifically “Global climate model simulations robustly indicate a slowdown of the AMOC in warmer future climate^{40–42}. These simulated slowdowns are associated with a combination of surface warming and freshening that reduce surface density⁴¹, increasing upper ocean stratification. We have emphasised here the importance of mixing between Atlantic and Arctic waters, as a substantial component of water mass transformation, necessary to sustain the lower limb and to

achieve the density and depth of the return flow. By inhibiting mixing through raised potential energy demand, stronger stratification will reduce the amount of turbulent kinetic energy available for this mixing. Reduced sub-surface mixing of Atlantic waters is thus potentially the key to overall AMOC slowdown, further reducing the depth to which surface overturned Atlantic waters are returned southward".

Reviewers' comments:

Reviewer #1 (Remarks to the Author):

I thank the authors for addressing my additional questions and comments. The paper has been improved by clarifying that the small contribution of Atlantic water transformation along the eastern periphery of the Nordic Seas to the lower limb of AMOC is valid for the trajectories that not entering Arctic Ocean. However, I am not convinced by the statement that “we showed that neither open ocean convection in the Greenland and Iceland Seas, mixing between gyre and boundary, are sufficient to transform waters to the high densities necessary to sustain the AMOC lower limb”. This is because the remaining 6.6 Sv Atlantic water that enters the Arctic Ocean can be transformed by lateral mixing with the gyre/interior water (include the indirect impact of open ocean convection) in the Nordic Seas. It also remains unclear regarding the role of Atlantic-Arctic mixing for the unconsidered 6.6 Sv Atlantic Water and 2.3 Sv Arctic Water (returns to the Arctic Ocean). The authors already have the remaining trajectories between the eastern SPNA and Fram Strait/Barents Sea before they were halted. What I expected is that the authors can do the same Lagrangian analysis (e.g., overturning streamfunction, divergence of heat/salt) for those remaining trajectories. This does not require any new trajectories in Arctic Ocean and can significantly fill in the gaps in the story.

One additional minor suggestion:

Every time I reviewed the Figure 3, it's difficult to tell the value (Sv) corresponding to each streamline, especially for the Arctic streamlines. Could the authors add some labels to better indicate streamline values in the Figure?

Reviewer #2 (Remarks to the Author):

Review of second revised version of “Formation of the Atlantic Meridional Overturning Circulation lower limb is critically dependent on Atlantic-Arctic mixing” by Dey et al. for Nature Communications

The authors have responded to the second review. There were three main points:

- (1) The choice of 43 years for the particle integration period is arbitrary, and the headline results depend on this choice.
- (2) The TRACMASS stream tube interpretation needs to be justified because there are reasons to think it may be unreliable.
- (3) The conclusion about the importance of air/sea interaction (rather than interior mixing) may be

overstated and needs to be justified.

Plus there was a minor fourth point about clarifying the mechanism of projected 21st century AMOC weakening.

Apart from one new paragraph on page 5 about new backtracking results (see below), and a few minor changes in the Discussion (see the tracked changes manuscript), the revised paper is essentially unchanged. Apart from the new Fig. S4, the supplement is also unchanged.

Therefore, I address the rebuttal to the previous review point-by-point:

The rebuttal to point (1) above (page 7 of rebuttal) avoids addressing the essential issue about sensitivity of the main results to integration time. The remark about the “the time-mean AMOC strength will not change significantly” isn’t relevant (point (1) remains true if the flow is steady or not). Most of the rest of the rebuttal to point (1) is about densification in the eastern Nordic Seas, which is also not relevant.

The rebuttal to point (2) is confusing. For example, the rebuttal says that the stream tube idea works for both steady and non-steady flows, although section 3.2.3 of van Sebille et al. (2018) says “the streamtube approach is formally only applicable” to steady-state velocity fields. Still, I think this is a minor point.

What is more confusing is: “Yes the flow is chaotic but we are making averages over large quantities of trajectories, which sums up as mean flows. Otherwise there would be no mean currents in the ocean transporting heat and salt in and between the oceans.” This doesn’t make sense to me. The point (reprising the previous reviews) is that Lagrangian trajectories are chaotic (in general), even for steady flow, and therefore particle trajectories diverge. This divergence of trajectories (not fluid) distorts the stream tube. The question is how long does it take for this distortion to occur, and is that long or short compared to 43 years. This question must be answered case-by-case. The rebuttal says that the proposed experiment to answer this question is “almost impossible”, however, because the code can’t handle it.

Nevertheless, the rebuttal partly addresses point (2) with new evidence. Specifically, results from a new particle experiment starting in 2021 and running backwards in time are shown. The stream function constructed using these results (new Fig. S4) resembles the one made with particles running forwards in time (Fig. 3c). This is a reassuring result. It shows that the basic idea of attributing a stream tube transport to each particle is reasonable, which is the essence of point (2) above. Still, the two stream functions aren’t identical: For example, the one in Fig. 3c shows inflow from the south at significantly higher densities than the one in Fig. S4. And the one in Fig. 3c shows inflow from the north at significantly lower densities than the one in Fig. S4. To clarify these differences, I suggest Figs 3a (temperature-latitude space) and 3b (salinity-latitude space) are also shown for the new backward integration. A balanced discussion of the similarities and differences between the forward and backward particle results, and the implications for the paper’s main claims, is also needed. This will guide the reader on the robust aspects of the overturning stream

function.

Currently, the statement in the Fig. S4 caption that the forward and backward stream functions are “strikingly similar” is perfunctory (the main text on page 5 says “very similar”, although the rebuttal on page 10 says “approximately the same”). Moreover, at the end of Results it says: “However, backward tracing is insufficient to quantify the total amount of transport from the Fram Strait and eSPNA sections, and respective partitioning, compared to the forward trajectories.” This is confusing. It seems to mean that the backward particle results show the stream tube idea works (“strikingly similar”), but they should be ignored (sentence above). This reasoning is superficial and uncritical.

The rebuttal to point (3) is also confusing. New evidence is required, which (presumably?) is “almost impossible” to acquire because of the code constraints. The rebuttal avoids engaging with the point, however, which is another loose end.

The final fourth point raised in the previous review about the mechanisms of projected AMOC weakening has been addressed fine.

For these reasons, overall and with regret, I still think the paper makes assertions that aren't fully supported by evidence. The rebuttal, in essence, declines to investigate these unsupported claims. I think the claims are probably right, but the detailed reasoning is shaky, and the details matter. Therefore, there's no choice but to continue recommending the manuscript is declined, and the authors are encouraged to carefully rethink their study and resubmit. The paper makes claims that are important, significant, and interesting; I just haven't yet seen enough detailed evidence to fully trust them.

Reply to reviewer 1 *Thank you for the response to our revised article. We are grateful for all your constructive suggestions, which have helped us to refine the manuscript. Below, we respond to each comment in turn. Line numbers are from the revised manuscript*

Comments

I thank the authors for addressing my additional questions and comments. The paper has been improved by clarifying that the small contribution of Atlantic water transformation along the eastern periphery of the Nordic Seas to the lower limb of AMOC is valid for the trajectories that not entering Arctic Ocean. However, I am not convinced by the statement that “we showed that neither open ocean convection in the Greenland and Iceland Seas, mixing between gyre and boundary, are sufficient to transform waters to the high densities necessary to sustain the AMOC lower limb”. This is because the remaining 6.6 Sv Atlantic water that enters the Arctic Ocean can be transformed by lateral mixing with the gyre/interior water (include the indirect impact of open ocean convection) in the Nordic Seas. It also remains unclear regarding the role of Atlantic-Arctic mixing for the unconsidered 6.6 Sv Atlantic Water and 2.3 Sv Arctic Water (returns to the Arctic Ocean). The authors already have the remaining trajectories between the eastern SPNA and Fram Strait/Barents Sea before they were halted. What I expected is that the authors can do the same Lagrangian analysis (e.g., overturning streamfunction, divergence of heat/salt) for those remaining trajectories. This does not require any new trajectories in Arctic Ocean and can significantly fill in the gaps in the story.

Answer : Thank you for raising an important point here. Just to clarify, the full sentence on line no: 168 - 171 was " We show that neither open ocean convection in the Greenland and Iceland Seas, mixing between gyre and boundary, nor heat loss of ocean waters along the eastern periphery of Nordic Seas (or in other words

along the NwAFC and Norwegian Atlantic Slope Current) which do not enter the Arctic Ocean are sufficient to transform waters to the high densities necessary to sustain the AMOC lower limb". We mentioned in our previous response and in the manuscript that it is possible that 6.6 Sv of Atlantic water entering the Arctic Ocean could be transformed in the Nordic Sea and Greenland Sea due to air-sea interaction and/or mixing. To explicitly prove this, and following your suggestion, we have now shown the stream function in tracer-latitude coordinates for water parcels which have entered the Arctic Ocean from the Atlantic Ocean in Supplementary Figure 6 and Supplementary Figure 7. We now state on line no: 171 - 174, "However, ocean heat loss along the Nordic Seas periphery can have an indirect effect on the Arctic outflow. After losing heat to the atmosphere, a substantial fraction of densified Atlantic waters travel further northward and enter the Arctic Ocean through Fram Strait (≈ 3 Sv) and the Barents Sea (≈ 3.3 Sv) as shown in Supplementary Fig.6,7".

One additional minor suggestion:

Every time I reviewed the Figure 3, it's difficult to tell the value (Sv) corresponding to each streamline, especially for the Arctic streamlines. Could the authors add some labels to better indicate streamline values in the Figure?

Answer : We realised that although we have mentioned the contour interval in Figure 3 caption, we did not specify the starting value of the contours. This might have caused the confusion. We have now updated the Figure 3 caption as "Meridional overturning streamfunction in a) temperature-latitude, b) salinity-latitude and c) density-latitude coordinate system. This has been computed using equation 3 and with trajectories that have exited the eastern SPNA section southward. The orange lines indicate Atlantic inflows and blue dashed lines correspond to Arctic waters that have reached the eastern SPNA. The direction of the water movement in each figure is indicated by arrows. The solid streamlines are 2 Sv apart and start at 1Sv. Similarly, the dashed streamlines are plotted with 1 Sv interval and

start at -1 Sv. The Atlantic and Arctic water contribution to the AMOC lower limb are shown with bold orange and blue numbers respectively". We have tried labelling the contours, but as the Atlantic contours are closely spaced the labels were unclear, and labelling only the Arctic streamlines would be inconsistent. We hope that by specifying the contour starting values and intervals in the figure caption, we have resolved this issue.

Reply to reviewer 2 *Thank you for the response to our revised article. We are grateful for all your constructive suggestions, which have helped us to refine the manuscript. Below, we respond to each comment in turn. Line numbers are from the revised manuscript*

Comments

The authors have responded to the second review. There were three main points: (1) The choice of 43 years for the particle integration period is arbitrary, and the headline results depend on this choice.

(2) The TRACMASS stream tube interpretation needs to be justified because there are reasons to think it may be unreliable.

(3) The conclusion about the importance of air/sea interaction (rather than interior mixing) may be overstated and needs to be justified.

The rebuttal to point (1) above (page 7 of rebuttal) avoids addressing the essential issue about sensitivity of the main results to integration time. The remark about the “the time-mean AMOC strength will not change significantly” isn’t relevant (point (1) remains true if the flow is steady or not). Most of the rest of the rebuttal to point (1) is about densification in the eastern Nordic Seas, which is also not relevant.

Answer : We have now added separately a new section “Dependency on chosen time period” on line no: 129 that describes in detail about the robust aspects of the current results when using different time periods.

The rebuttal to point (2) is confusing. For example, the rebuttal says that the stream tube idea works for both steady and non-steady flows, although section 3.2.3 of van Sebille et al. (2018) says “the streamtube approach is formally only applicable” to steady-state velocity fields. Still, I think this is a minor point.

Answer : The method we use has its origin in applications with steady-state velocity fields, and the streamtube approach is formally only applicable to these

cases. However, by sub-sampling these fields into piecewise steady or stepwise-stationary fields, the streamtube approach can also be used for time-varying flow. Section 3.2.3 of van Sebille et al. (2018) also states that “While these methods have their origin in applications with steady-state velocity fields, and the streamtube approach is formally only applicable to these cases, there is a large community using extended analytical discrete streamtube methods for time-varying flows too”. As mentioned in the earlier response we have used time-evolving fields and sub-sampled them into piecewise steady or stepwise-stationary fields. This is also mentioned in Section 3.2.3 of van Sebille et al. (2018), which states that “If the velocity fields are time-evolving, it is possible to sub-sample them into piecewise steady fields, which are only kept constant in time for a short time; this approach generally increases the computational cost”. Line no: 212 - 215 of the manuscript also mentioned this as “For time-dependent mass/volume transport fields, TRACMASS simulates trajectory routes with an analytical solution by solving a differential equation that is based on linear interpolation of mass/volume transports on the model grid box walls. This scheme is known as stepwise-stationary, in which mass/volume transport fields are assumed to be stationary during intermediate time steps between two model output sets”.

What is more confusing is: “Yes the flow is chaotic but we are making averages over large quantities of trajectories, which sums up as mean flows. Otherwise there would be no mean currents in the ocean transporting heat and salt in and between the oceans.” This doesn’t make sense to me. The point (reprising the previous reviews) is that Lagrangian trajectories are chaotic (in general), even for steady flow, and therefore particle trajectories diverge. This divergence of trajectories (not fluid) distorts the stream tube. The question is how long does it take for this distortion to occur, and is that long or short compared to 43 years. This question must be answered case-by-case. The rebuttal says that the proposed experiment to answer this question is “almost impossible”, however, because the code can’t handle it.

Answer : The concept of streamtube applies to every single trajectory and each trajectory carries a certain amount of volume transport which is conserved. First all these single streamtubes are close together with no other watermass in between them like the veins in an arm, but after some distance the arms/hand splits in separate fingers and after some distance each finger splits in various tentacles etc. But the number of streamtubes (= number of trajectories = volume transport) remains constant, although they diverge more and more. Thus although the fluid (here represented by trajectories) is chaotic, it still sums up and gives rise to a mean flow. To represent a mean realistic flow, sufficient number of trajectories are required. In the present study we are making averages over large quantities of trajectories, which sums up as mean flows. This approach has been used in numerous studies over 30 years e.g., Döös et al.,2008, Berglund et al.,2023 etc. to show that if we have sufficient number of trajectories then the Eulerian mean and Lagrangian mean fields are the same. To check if a forward trajectory and backward trajectory take the same path, we mentioned in our earlier response that it is “almost impossible” to check using real data as the code cannot start from an interpolated time step. We were wrong to say TRACMASS cannot handle it instead it is data unavailability at interpolated time step from model/reanalysis. We gave an example in earlier response and we wanted to reiterate here. Suppose that a trajectory starts at the Fram Strait in January 1979 (we can do this as we have monthly data from the hindcast model) and reaches the eSPNA section after 11.3 years. Although we know the ending position of that trajectory, we cannot start a backward experiment from the same location, as the hindcast model (or any model/reanalysis) does not provide data at an interpolated time step (in the above example, we would need data on day 18.25 of March 1990). In principle, a TRACMASS forward/backward trajectory will however be the same if we start and end the two trajectories at the same position in time and space. Appendix A1.2 of van Sebille et al. (2018) states that “TRACMASS solves the path of a trajectory through a grid box analytically (see Section 3.2.3) Trajectories are thus unique and if a trajectory is calculated forward and then backward the solution

will be the same up to numerical noise due to round-off errors”.

Döös, Kristofer and Nycander, Jonas and Coward, Andrew C. Lagrangian decomposition of the Deacon Cell. Journal of Geophysical Research: Oceans. 113(C7), (2008)

Berglund, Sara and Döös, Kristofer and Groeskamp, Sjoerd and McDougall, Trevor. North Atlantic Ocean circulation and related exchange of heat and salt between water masses. Geophysical Research Letters. 50(13), e2022GL100989, (2023)

Nevertheless, the rebuttal partly addresses point (2) with new evidence. Specifically, results from a new particle experiment starting in 2021 and running backwards in time are shown. The stream function constructed using these results (new Fig. S4) resembles the one made with particles running forwards in time (Fig. 3c). This is a reassuring result. It shows that the basic idea of attributing a stream tube transport to each particle is reasonable, which is the essence of point (2) above. Still, the two stream functions aren't identical: For example, the one in Fig. 3c shows inflow from the south at significantly higher densities than the one in Fig. S4. And the one in Fig. 3c shows inflow from the north at significantly lower densities than the one in Fig. S4. To clarify these differences, I suggest Figs 3a (temperature-latitude space) and 3b (salinity-latitude space) are also shown for the new backward integration. A balanced discussion of the similarities and differences between the forward and backward particle results, and the implications for the paper's main claims, is also needed This will guide the reader on the robust aspects of the overturning stream function.

Answer : Thank you for this constructive suggestion. We have now added a new section in the revised manuscript named “Dependency on chosen time period” on line no: 129. This section discusses the similarities and differences between the forward and backward trajectory simulated results. The temperature-latitude

and salinity-latitude stream functions are now calculated from the backward simulation and provided in Supplementary Fig. 4a,4b. In a nutshell, due to time variability forward and backward tracing will never give exactly the same results but that differences are acceptable and do not change our conclusions.

Currently, the statement in the Fig. S4 caption that the forward and backward stream functions are “strikingly similar” is perfunctory (the main text on page 5 says “very similar”, although the rebuttal on page 10 says “approximately the same”). Moreover, at the end of Results it says: “However, backward tracing is insufficient to quantify the total amount of transport from the Fram Strait and eSPNA sections, and respective partitioning, compared to the forward trajectories.” This is confusing. It seems to mean that the backward particle results show the stream tube idea works (“strikingly similar”), but they should be ignored (sentence above). This reasoning is superficial and uncritical.

Answer : We have now removed the words “strikingly similar”, “very similar”, “approximately the same” from the main texts and also modified the Supplementary Figure 4 caption as “Meridional overturning streamfunction in a) temperature-latitude, b) salinity- latitude and c) density-latitude coordinate system. This has been computed with backward trajectories that have started and exited at the eastern SPNA section. The orange lines indicate Atlantic inflows and blue dashed lines correspond to Arctic waters that have reached the eastern SPNA. The direction of the water movement in each figure is indicated by arrows. The contour intervals are 2 Sv and start at 1 Sv for solid contours. The dashed streamlines are plotted with 1 Sv interval and start at -1 Sv. The Atlantic and Arctic water contributions to the AMOC lower limb are shown with bold orange and blue numbers respectively. Note, the streamfunctions closely resemble those in Figure 3c, which were obtained using the forward trajectory simulation.” The example figure (below) explains that in Forward trajectory run total transport from the Fram strait and eSPNA section is obtainable. However, backward track-

ing of only southward flow across eSPNA section will not provide Fram strait to strait, Fram strait to Barents Sea, northward flow from eSPNA to Fram strait and eSPNA to Barents Sea estimates. This is what we meant previously by saying “However, backward tracing is insufficient to quantify the total amount of transport from the Fram Strait and eSPNA sections, and respective partitioning, compared to the forward trajectories”. In the backward trajectory experiment, we can only obtain eSPNA to eSPNA circulating transports and Arctic Ocean to eSPNA section transports as these will only be southward moving across eSPNA.

Example Figure 1: A sketch to illustrate the similarities and dissimilarities between forward and backward trajectory experiments. The cross symbol indicates what we cannot obtain from the backward trajectory experiment.

The rebuttal to point (3) is also confusing. New evidence is required,

which (presumably?) is “almost impossible” to acquire because of the code constraints. The rebuttal avoids engaging with the point, however, which is another loose end.

Answer : To check if a forward trajectory and backward trajectory takes the same path, we mentioned in our earlier response that it is “almost impossible” to check using real data as the code cannot start from an interpolated time step. We were wrong to say TRACMASS cannot handle it; rather we are limited by data unavailability at interpolated time steps in the model/reanalysis. We gave an example in the earlier response and reiterate it here. Suppose that a trajectory starts at the Fram Strait in January 1979 (we can do this as we have monthly data from the hindcast model) and reaches the eSPNA section after 11.3 years. Although we know the ending position of that trajectory, we cannot start a backward experiment from the same location, as the hindcast model (or any model/reanalysis) does not provide data at an interpolated time step (in the above example, we would need data on day 18.25 of March 1990).

REVIEWER COMMENTS

Reviewer #2 (Remarks to the Author):

The authors have revised their manuscript in response to a third round of reviews, which contain three main points (see below). There's a new subsection called "Dependency on chosen time period", two new supplementary figures (S6, S7), and a few other updates. The authors also give detailed text responses to the last review. I comment on the responses to the three main points of the earlier reviews, which are:

- (1) The choice of 43 years for the particle integration period is arbitrary, and the headline results depend on this choice.
- (2) The TRACMASS stream tube interpretation needs to be justified because there are reasons to think it may be unreliable.
- (3) The conclusion about the importance of air/sea interaction (rather than interior mixing) may be overstated and needs to be justified.

The response to (1) is the new subsection "Dependency on chosen time period". This text talks about how the Atlantic/Arctic contributions to the AMOC lower limb depend on the chosen time period in the following sense: the contributions in (say) the 1990s can be different to the contributions in the 2010s. This is correct because the currents (and mixing) vary between the 1990s and 2010s. However, it doesn't address point (1). Point (1) is about the choice of 43 years, not the secular change in the currents. This is explained in the 1st and 2nd reviews.

The revised figure S4 (backtracking streamfunction results) is now in good shape and allows a careful comparison with figure 3. They're quite similar, which is reassuring. The 2nd paragraph in "Dependency on chosen time period" discusses these results in a reasonable way. I have 1 minor comment: The differences between figures 3 and S4 is attributed by the authors to secular change, specifically it's warmer in the SPNA in 2016 than in 1984. This is plausible (and easily checked), but it's not the only possibility. An alternative explanation is that the forward and backward trajectories don't exactly coincide. I made this point in earlier reviews.

To resolve point (1) the authors should add text to "Dependency on chosen time period" to (i) explain how the results depend on the choice of 43 years, rather than any other period (see the previous reviews) and (ii) explain that the differences between figures 3 and S4 might also be due to mismatch between the forward and backward trajectories (see previous reviews and below).

The response to point (2) is a long, useful discussion of the TRACMASS Lagrangian tool. I have 1 comment on this response at page 6: The text says "In principle, a TRACMASS forward/backward trajectory will...be the same if we start and end the two trajectories at the same position in time and space." I agree that this is true in principle, however, it is false in practice. The reason is the

“numerical noise due to round-off errors” (top of page 7 in rebuttal, quoted from van Sebille et al. 2018). These errors will (in general) cause exponential divergence of nearby trajectories (and of forward and backward trajectories) due to the Lagrangian chaos explained in the previous reviews. Using an analytic trajectory scheme in TRACMASS doesn’t avoid the chaotic divergence of trajectories and therefore the eventual destruction of a stream tube. As stated before, the question is how long does that take. Anyway, this is a technical point. The fact that figures 3 and S4 look similar suggests that 43 years isn’t long compared to this divergence timescale (the forward and backward trajectories mainly coincide).

The response to point (3) clarifies why the authors can’t check the claim that 67% of Atlantic waters are transformed by air-sea interaction. The 67% number is an important result (2nd paragraph of discussion), however. So if the authors can’t check it, main body text explaining the uncertainty is essential. The 67% assertion is plausible, but not the only possibility (see earlier reviews). The reader must know this.

Reply to reviewer 2 *Thank you for the response to our revised article. We are grateful for all your constructive suggestions, which have helped us to improve the manuscript. Below, we respond to each comment in turn. Line numbers are from the revised manuscript.*

Comments

We comment on the responses to the three main points of the earlier reviews, which are:

(1) The choice of 43 years for the particle integration period is arbitrary, and the headline results depend on this choice.

(2) The TRACMASS stream tube interpretation needs to be justified because there are reasons to think it may be unreliable.

(3) The conclusion about the importance of air/sea interaction (rather than interior mixing) may be overstated and needs to be justified.

The response to (1) is the new subsection "Dependency on chosen time period". This text talks about how the Atlantic/Arctic contributions to the AMOC lower limb depend on the chosen time period in the following sense: the contributions in (say) the 1990s can be different to the contributions in the 2010s. This is correct because the currents (and mixing) vary between the 1990s and 2010s. However, it doesn't address point (1). Point (1) is about the choice of 43 years, not the secular change in the currents. This is explained in the 1st and 2nd reviews.

Thank you for your relentless effort to help us to improve the quality of the manuscript. Please find our responses below.

The revised figure S4 (backtracking streamfunction results) is now in good shape and allows a careful comparison with figure 3. They're quite similar, which is reassuring. The 2nd paragraph in "Dependency on chosen time period" discusses these results in a reasonable way. I have 1

minor comment: The differences between figures 3 and S4 is attributed by the authors to secular change, specifically it's warmer in the SPNA in 2016 than in 1984. This is plausible (and easily checked), but it's not the only possibility. An alternative explanation is that the forward and backward trajectories don't exactly coincide. I made this point in earlier reviews.

To resolve point (1) the authors should add text to "Dependency on chosen time period" to (i) explain how the results depend on the choice of 43 years, rather than any other period (see the previous reviews) and (ii) explain that the differences between figures 3 and S4 might also be due to mismatch between the forward and backward trajectories (see previous reviews and below).

Answer : Thank you for the constructive comments. Following your suggestion we have now discussed the dependency of results on the choice of 43 years and explained the differences between figures 3 and S4 on line no: 164 - 170 as "The choice of 43 years for trajectory simulations was done to ensure that most trajectories (or volume transport) have had time to exit the study domain (Supplementary Fig.5). When it comes to the small difference of the stream functions in Figure 3 and Supplementary Fig.4 due to mismatch in forward and backward trajectory pathways, it is important to note that the trajectories will not disperse evenly in space but be concentrated where the strong currents are located e.g., the East Greenland Current, Norwegian Atlantic Slope Current etc (Figure 2a-b) The similar shape of stream functions obtained from the forward and backward trajectory (Figure 3 and Supplementary Fig.4) experiments also validate this argument that trajectories mainly follow the strong mean currents" We realised, however, that it is not easy to connect Lagrangian statistics (relative dispersion, Lagrangian chaotic advection timescale etc.) with mass transport studies using stream tubes such as in the present study. It would be interesting to study and explain this in a separate article, where we clarify how it is possible to track masses of water in the ocean and air in the atmosphere despite its chaotic nature.

The response to point (2) is a long, useful discussion of the TRACMASS Lagrangian tool. I have 1 comment on this response at page 6: The text says "In principle, a TRACMASS forward/backward trajectory will...be the same if we start and end the two trajectories at the same position in time and space." I agree that this is true in principle, however, it is false in practice. The reason is the "numerical noise due to round-off errors" (top of page 7 in rebuttal, quoted from van Sebille et al. 2018). These errors will (in general) cause exponential divergence of nearby trajectories (and of forward and backward trajectories) due to the Lagrangian chaos explained in the previous reviews. Using an analytic trajectory scheme in TRACMASS doesn't avoid the chaotic divergence of trajectories and therefore the eventual destruction of a stream tube. As stated before, the question is how long does that take. Anyway, this is a technical point. The fact that figures 3 and S4 look similar suggests that 43 years isn't long compared to this divergence timescale (the forward and backward trajectories mainly coincide).

Answer : Thank you for the comment. Your thoughts are principally correct, but in practice these "errors" or uncertainties are very small, much smaller than the model error in simulating the flow field used for the Lagrangian Diagnosis. The choice of 43 years and the mismatch in forward and backward trajectory pathways are already discussed in the previous response. Additionally, we have previously performed a sensitivity analysis where we changed the maximum volume transport assigned to each trajectory by $\pm 500 \text{ m}^3\text{s}^{-1}$. We realised that by doing so we were not only changing the amount of trajectories but also their starting positions inside a grid box. In these perturbed simulations the trajectories as you have stated will disperse differently. However, they still produce very similar results (within 1% uncertainty level). This affirms the fact that the trajectories primarily follow the strong mean currents. The sensitivity experiment is already mentioned on line no: 254 - 256.

The response to point (3) clarifies why the authors can't check the claim that 67% of Atlantic waters are transformed by air-sea interaction. The 67% number is an important result (2nd paragraph of discussion), however. So if the authors can't check it, main body text explaining the uncertainty is essential. The 67% assertion is plausible, but not the only possibility (see earlier reviews). The reader must know this.

Answer : Thank you for the comment. A part of the response related to this query has been provided in earlier statements. We have now stated on line no: 152 - 154 that "Backward tracing reveals that the Atlantic and Arctic contribution to the AMOC lower limb have uncertainty of 3% as compared to the forward trajectory simulation". Furthermore, we have mentioned on line no: 158 - 164 that "The noted differences in the stream functions obtained from forward and backward Lagrangian tracking might also be associated with the inability to start trajectories from their exact ending position after a forward/backward tracking simulation using time-evolving fields. This is since we are limited by data unavailability at interpolated time steps in the model/reanalysis. Consider for example a trajectory that starts at the Fram Strait in January 1979 (using monthly data from the hindcast model) and reaches the eSPNA section after 11.3 years. Although we know the ending position of that trajectory, we cannot start a backward run from the same location, as the hindcast model (or any model/reanalysis) does not provide data at an interpolated time step (in the above example, we would need data on day 18.25 of March 1990)." In the section "Dependency on chosen time period" we emphasise that the results obtained with forward and backward trajectories are not exactly the same and discuss the reasons for this. Given our explanation, we anticipate that readers will interpret our results –accurate to within a few percent – as clearly indicative of the relative importance of air-sea interactive and mixing.

REVIEWERS' COMMENTS

Reviewer #2 (Remarks to the Author):

The responses to points (1) and (2) from the previous review are fine.

The response to point (3) about the 67% of Atlantic waters are transformed by air-sea interaction is not relevant. As stated before, if the authors want to quote 67% (lines 112 and 178), they should state in the main text that they can't check it, although it's a plausible value.

Reply to reviewer 2 *Thank you for the response to our revised article. We are grateful for all your constructive suggestions, which have helped us to improve the manuscript. Below, we respond to each comment in turn. Line numbers are from the revised manuscript.*

Comments

The responses to points (1) and (2) from the previous review are fine.

Answer : Thank you.

The response to point (3) about the 67% of Atlantic waters are transformed by air-sea interaction is not relevant. As stated before, if the authors want to quote 67% (lines 112 and 178), they should state in the main text that they can't check it, although it's a plausible value.

Answer : Thank you for the comment. We have now mentioned on line number 156 - 158 that "Based on our explanation, it is important to note that the estimate of 67% of Atlantic waters overturning due to air-sea fluxes and 33% related to interior mixing is accurate to within a few percent, clearly highlighting the relative importance of air-sea interactions and mixing processes."